# AGENT-ENVIRONMENT ALIGNMENT VIA AUTOMATED INTERFACE GENERATION

## ABSTRACT

Large language model (LLM) agents have shown impressive reasoning capabilities in interactive decision-making tasks. These agents interact with environment through intermediate interfaces, such as predefined action spaces and interaction rules, which mediate the perception and action. However, mismatches often happen between the internal expectations of the agent regarding the influence of its issued actions and the actual state transitions in the environment, a phenomenon referred to as **agent-environment misalignment**. While prior work has invested substantially in improving agent strategies and environment design, the critical role of the interface still remains underexplored. In this work, we empirically demonstrate that agent-environment misalignment poses a significant bottleneck to agent performance. To mitigate this issue, we propose **ALIGN**, an Auto-Aligned Interface Generation framework that alleviates the misalignment by enriching the interface. Specifically, the ALIGN-generated interface enhances both the static information of the environment and the step-wise observations returned to the agent. Implemented as a lightweight wrapper, this interface achieves the alignment without modifying either the agent logic or the environment code. Experiments across multiple domains including embodied tasks, web navigation and tool-use, achieve consistent performance improvements, with up to a 45.67% success rate improvement observed in ALFWorld. Meanwhile, ALIGN-generated interface can generalize across different agent architectures and LLM backbones without interface regeneration.

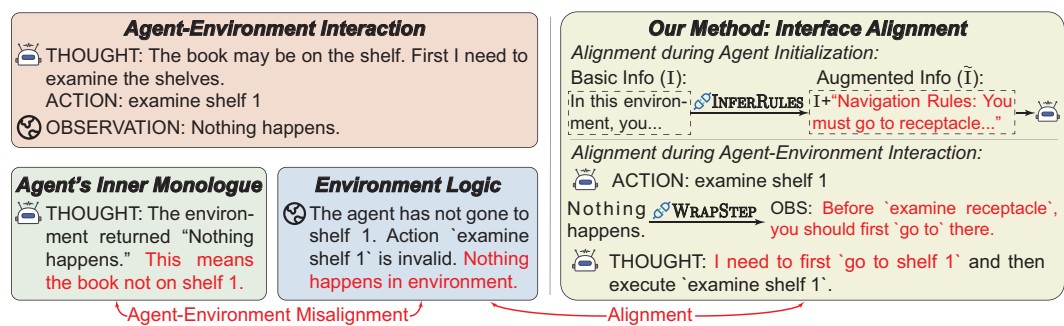

Figure 1: **Illustration of agent-environment misalignment and our proposed solution. Left:** The agent and the environment have a misalignment in their interpretation of the same observation, where the agent's understanding of the observation differs from the environment's underlying logic. **Right:** ALIGN bridges this gap via an automatically generated interface comprising two modules, **INFERRULES** and **WRAPSTEP**: (1) During initialization, INFERRULES augments the basic information with explicit environment constraints. (2) During interaction, WRAPSTEP translates the raw observation into an informative observation that better conveys the environment state transitions.

## 1 INTRODUCTION

Large Language Model (LLM) agents have demonstrated promising performance in interactive tasks such as embodied tasks (Driess et al., 2023; Lin et al., 2023; Wang et al., 2024a), web

navigation tasks (Chae et al., 2025; He et al., 2024a; Qi et al., 2024), and tool-use tasks (Wang et al., 2024b; Paranjape et al., 2023; Schick et al., 2023). In these tasks, **agents** typically interact with the **environment** through manually designed **interfaces** such as predefined action spaces and interaction rules. While substantial efforts have been devoted to improving agents and environments, comparatively little attention has been paid to the interface between them, leading to a problem we term **agent-environment misalignment**, which significantly impacts the agent performance.

The agent-environment misalignment refers to the discrepancy between the interpretation of the agent to the observation following an action and the underlying logic of the environment. As illustrated in Figure 1 (left), in ALFWorld (Shridhar et al., 2021), issuing *examine receptacle* fails unless the agent first executes "`go to receptacle`". Consequently, the environment responds with the observation "Nothing happens.". At this point, the agent interprets the observation to mean that there is nothing on shelf 1, which is inconsistent with the underlying reason for the environment providing it. To assess the impact of this misalignment, we conduct preliminary experiments, which reveal that simply revising the observation for an invalid "`examine receptacle`" action to "You need to first go to receptacle before you can examine it" increases the success rate of a vanilla Qwen2.5-7B-Instruct (Team, 2024) agent on ALFWorld from 13.4% to 31.3%[1]. Such phenomenon suggests that the agent-environment misalignment significantly hinders task success, and can be alleviated by improving interface design. From the perspective of the agent, poorly designed interfaces impose unnecessary cognitive overhead. Furthermore, from an evaluation perspective, inadequate interfaces can obscure an accurate assessment of the true reasoning capabilities of agents. Therefore, we argue that the problem of agent-environment misalignment warrants greater attention.

However, addressing the agent-environment misalignment is challenging. On one hand, current benchmarks primarily focus on advance agent intelligence by constructing increasingly complex and challenging environments (Jimenez et al., 2024; Wang et al., 2025b; Wei et al., 2025; Xie et al., 2024; Zhou et al., 2024a), often overlooking the importance of improving interface design. This oversight extends across multiple domains of interactive tasks, such as, ALFWorld, OSWorld (Xie et al., 2024), and M³ToolEval (Wang et al., 2024b). They all exhibit similar deficiencies: failing to provide agent-parseable observations for environmental constraints violation in embodied tasks, positional inaccuracies in operating system tasks or parameter format errors in multi-turn tool-use tasks, respectively. On the other hand, although some recent work (Agashe et al., 2024; Yang et al., 2024a; Zheng et al., 2024) has begun to consider interface design, these efforts often rely on manual, environment-specific tailoring, which introduces two critical issues: (1) they are highly labor-intensive and (2) whether human-designed interfaces are optimal for agents remains an open question.

Furthermore, in addition to studies that explicitly optimize interface design, it is common in agent-focused research for researchers to manually re-engineer environment interfaces to align with their specific methods. For instance, for the same environment ALFWorld, Zhou et al. (2024b) manually maintains the environment's state information in JSON format; Ma et al. (2024) introduces a new action *check_valid_actions* to enable agents to retrieve all valid actions; and Chen et al. (2024a) re-implements the environment by wrapping it into a new class *InteractEnv*. However, such ad-hoc customization pose a significant challenge to the field: it compromises the direct comparability across different approaches. Moreover, these modifications are often tailored to the specific methods proposed, making it difficult for the research community to determine whether performance variations stem from novel agent architectures or from the non-standardized, customized interfaces. Therefore, we believe that manually re-engineering environment interfaces is not an optimal approach to alleviating the agent-environment misalignment problem.

Distinct from the aforementioned works, we propose to **automatically generate interfaces for bridging the agent-environment misalignment**. In this work, we introduce **ALIGN** (Auto-Aligned Interface Generation), a framework that automatically generate aligned interfaces for environments. The generated interface consists of two modules: INFERRULES and WRAPSTEP. The former automatically discovers and provides the agent with static information about environmental rules or internal constraints, facilitating *static alignment*, while the latter enhances the interaction by offering more detailed observations for agent-issuing actions, enabling *dynamic alignment*, as shown in Figure 1 (right). Owing to the powerful reasoning and coding capabilities of current advanced LLMs, we utilize these models to analyze existing agent-environment misalignments and automatically generate the interface. Moreover, we employ LLMs to conduct experimental verification to mitigate

---

[1]Experimental details are provided in Appendix C.1.

hallucination issues (Bang et al., 2023; Xu et al., 2024). Specifically, our LLM-based system autonomously validate both proposed misalignments and generated interface through direct interaction with the environment, ensuring that identified issues genuinely exist and are properly addressed by the interface. The generated interface acts as a lightweight wrapper, providing richer context and explicit constraint hints, enabling different LLM agents to align with the environment directly.

To evaluate the effectiveness of ALIGN, we conduct experiments on four representative benchmarks across three domains: embodied tasks, web navigation, and tool-use tasks. Our results demonstrate consistent performance improvements across all four benchmarks when using the ALIGN-generated interface, with notably gains of 45.67% in average success rate on ALFWorld. Moreover, the performance of GPT-4.1-based agents on ALFWorld are improved from 73.88% to 93.28% with ALIGN, highlighting the efficiency of our approach in mitigating the agent-environment misalignment to unleash the agent's true capabilities.

Our key contributions can be summarized as follows:

- We identify and characterize the **agent-environment misalignment** problem, empirically showing its prevalence across diverse domains and its role as a key bottleneck to agent performance.
- We introduce **ALIGN**, the first framework that automatically generates aligned interfaces to alleviate agent-environment misalignment, without modifying agent logic or environment code.
- We demonstrate the effectiveness and generalizability of **ALIGN** across three domains, with up to a 45.67% success rate improvement on ALFWorld.

## 2 RELATED WORK

**Agent-environment interface** The agent-environment interface defines how agents interact with the environment. In reinforcement learning, researchers construct unified interaction interfaces (Bonnet et al., 2024; Brockman et al., 2016; Kolve et al., 2017; Towers et al., 2024) to standardize the application and evaluation of different algorithms. With the increasing capability of LLMs to perform human-like actions (Guo et al., 2024; Liu et al., 2024; Ma et al., 2024), interface design has been proven to largely influence the performance of LLM-based agents (Xie et al., 2024; Rawles et al., 2024). SWE-agent (Yang et al., 2024a) proposes agent-computer interfaces for coding agents and recent efforts aim to improve generalization (Agashe et al., 2024; Qin et al., 2025; Niu et al., 2024) and enhance interfaces with auxiliary tools (Bula et al., 2025; Gou et al., 2024; Lei et al., 2025; Lu et al., 2024; Yang et al., 2023a). Nevertheless, current agent-environment interfaces are mostly manually crafted and tailored for specific environments or agent frameworks, limiting their generalization and scalability. Therefore, we propose automated interface generation to empower agents with effective, generalizable and automatic interface alignment.

**Methods aligning agents with environments** LLM agents have exhibited strong potential for real-world interaction and task completion Yao et al. (2023); Shinn et al. (2023); Liu et al. (2024). Current research in this area can be broadly categorized into training-based methods and training-free methods. Training-based methods consists of fine-tuning LLMs with expert-level interaction trajectories Zeng et al. (2024); Chen et al. (2023; 2025); Fu et al. (2025); Chen et al. (2024b) and enhancing environment-aligned planning and acting via reinforcement learning Bai et al. (2025); Yang et al. (2024b); Qi et al. (2024); Feng et al. (2024); Zhou et al. (2024c); Wang et al. (2025a). Though effective, these methods suffer from high computational costs and limited generalization towards unseen environments. Another approach constructs training-free multi-agent frameworks for task decomposition and experience accumulation (Chen et al., 2024a; He et al., 2024b; Sun et al., 2024; Yang et al., 2023b; Zhou et al., 2024b). However, static agent pipelines lack flexibility and experience injected through prompting often fails to capture environment dynamics and is not effectively utilized by LLMs, resulting in insufficient alignment between agents and environments.

## 3 METHOD

### 3.1 PROBLEM FORMULATION

**Environment and Agent.** In interactive decision-making tasks, we define the environment $\mathcal{E}$ as a tuple $(\mathcal{S}, \mathcal{A}, T, F, \mathcal{I})$, where $\mathcal{S}$ denotes the set of all possible states of the environment; $\mathcal{A}$ denotes the set of actions the agent can invoke; $T : \mathcal{S} \times \mathcal{A} \rightarrow \mathcal{S}$ defines how the environment state evolves in

response to agent actions; $F : \mathcal{S} \times \mathcal{A} \rightarrow \mathcal{O}$ provides textual feedback that reflects the consequences of the action in the current state, where $\mathcal{O}$ is all possible observations; $\mathcal{I}$ encodes the *environment foundational information description*, a fixed, declarative representation of the environment's basic introduction, which is exposed to the agent at initialization.

An agent $\pi$ interacts with environment $\mathcal{E}$ by receiving task description and observations, then producing actions $a_t \in \mathcal{A}$. The interaction generates trajectory $\tau = [(s_0, a_0, o_0), \ldots, (s_T, a_T, o_T)]$, culminating in task completion feedback.

**Formal Definition of Interface.** Existing works typically assume the agent interacts directly with $\mathcal{E}$. However, we argue that this interaction is mediated by an **Interface**, denoted as $\Phi$, which acts as a translation layer between the agent's cognitive space and the environment's execution space. Formally, we define the interface as a tuple of mapping functions: $\Phi = \langle f_{\text{info}}, f_{\text{act}}, f_{\text{obs}} \rangle$ where each component serves a distinct role:

- **Information Augmenter** $f_{\text{info}} : \mathcal{I} \rightarrow \tilde{\mathcal{I}}$ exposes implicit environment logic (e.g., constraints, admissible action sequences) into an explicit descriptive context $\tilde{\mathcal{I}}$ provided at agent initialization.
- **Action Transducer** $f_{\text{act}} : \mathcal{A}_{\text{agent}} \rightarrow \mathcal{A}_{\text{env}} \cup \{\bot\}$ maps the agent's output to an executable environment command. If the output cannot be parsed, it returns an invalid symbol $\bot$.
- **Observation Transducer** $f_{\text{obs}} : \mathcal{S} \times \mathcal{A}_{\text{env}} \times \mathcal{O}_{\text{raw}} \rightarrow \mathcal{O}_{\text{agent}}$ transforms the raw feedback $\mathcal{O}_{\text{raw}}$ (from $F$) into an informative observation $\mathcal{O}_{\text{agent}}$ that better conveys the actual state transitions and their causes.

At each timestep $t$, the agent receives $\tilde{o}_t \in \mathcal{O}_{\text{agent}}$ (processed by $f_{\text{obs}}$) and generates $a_t \in \mathcal{A}_{\text{agent}}$, which is then executed as $a_t^{\text{env}} = f_{\text{act}}(a_t) \in \mathcal{A}_{\text{env}}$.

**Agent-Environment Misalignment.** We analyze the misalignment problem through the lens of the interface $\Phi$. Ideally, $\Phi$ should be *lossless*, maximizing the mutual information between the agent's belief state and the ground-truth environment state. However, manually designed interfaces often exhibit **Semantic Gaps**, leading to misalignment through two primary mechanisms:

- **State Aliasing via Lossy Observations** ($f_{\text{obs}}$): A poorly designed $f_{\text{obs}}$ may map distinct error states (e.g., "action invalid due to wrong location" vs. "action invalid due to missing object") to the same generic observation (e.g., "Nothing happens."). This creates state aliasing, preventing the agent from diagnosing failures and correcting its policy.
- **Under-specified Constraints** ($f_{\text{info}}$): When critical transitions $T$ rely on preconditions (e.g., `open` requires `go to` first) that are not explicitly encoded in $\tilde{\mathcal{I}}$ by $f_{\text{info}}$, the agent operates under a hallucinated world model where such constraints appear absent.

Therefore, we define **Agent-Environment Misalignment** as the discrepancy between the agent's expected state transition $s_{t+1}^{\text{expected}}$ (derived from its internal world model based on $\tilde{\mathcal{I}}$ and prior observations $[\tilde{o}_0, \tilde{o}_1, \ldots, \tilde{o}_{t+1}]$) and the actual transition $s_{t+1}^{\text{actual}} = T(s_t, a_t^{\text{env}})$, caused by insufficient expressiveness of the interface $\Phi$.

**Scope of This Work.** While a complete interface theoretically includes all three components, we observe that misalignment in existing benchmarks primarily stems from information loss in $f_{\text{info}}$ and $f_{\text{obs}}$, rather than from action space incompatibility. Therefore, ALIGN focuses on *automatically optimizing* these two components, treating $f_{\text{act}}$ as a fixed identity mapping throughout this work: $f_{\text{act}}(a) = a$. This design choice allows us to address the core misalignment issues without modifying the agent's action generation logic or the environment's execution layer.

## 3.2 ALIGN OVERVIEW

To alleviate agent-environment misalignment, we introduce **ALIGN** (Auto-Aligned Interface Generation), a framework that automatically generates an optimized interface $\Phi^*$ to bridge the semantic gaps identified in Section 3.1. Specifically, ALIGN focuses on learning improved $f_{\text{info}}$ and $f_{\text{obs}}$ functions that minimize information loss during agent-environment interaction.

**Interface Instantiation.** As illustrated in Figure 2, ALIGN instantiates the theoretical interface components through two learnable modules implemented as a lightweight Python wrapper, without modifying the environment code or agent logic.

**INFERRULES** (implements $f_{\text{info}}$):  Transforms raw environment information $\mathcal{I}$ into augmented information $\tilde{\mathcal{I}}$ that explicitly exposes interaction rules and constraints. Formally: INFERRULES : $(\text{task}, o_0, \mathcal{I}) \rightarrow \tilde{\mathcal{I}}$, where $\tilde{\mathcal{I}}$ includes the constraints automatically extracted, such as precondition dependencies or action ordering requirements.

**WRAPSTEP** (implements $f_{\text{obs}}$):  Intercepts the raw observation function $F$ and augments its output to resolve state aliasing. Given the current state $s_t$ and agent action $a_t$, formally: WRAPSTEP : $(F, s_t, a_t) \rightarrow \tilde{o}_t$, where $\tilde{o}_t$ encapsulates both $F(s_t, a_t)$ and additional diagnostic or corrective information.

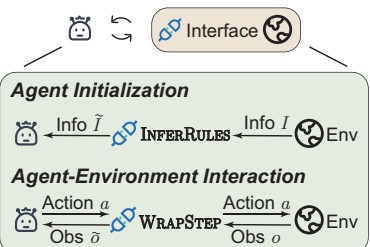

Figure 2: Overview of the ALIGN-generated interface. The interface wraps the original environment $\mathcal{E}$ to create an augmented environment $\tilde{\mathcal{E}}$. INFERRULES enriches static information ($\mathcal{I} \rightarrow \tilde{\mathcal{I}}$) at agent initialization, while WRAPSTEP augments step-wise observations ($F \rightarrow \tilde{F}$) during interaction.

Together, these modules form an intermediate interface wrapper layer that intercepts and transforms environment information before it reaches the agent. This design allows the base agent $\pi$ to remain unchanged, while still benefiting from contextual clarity and enriched observation that help avoid misaligned actions. From the perspective of the agent, interaction now occurs with an *augmented environment*, denoted as $\tilde{\mathcal{E}} = (\mathcal{S}, \mathcal{A}, T, \tilde{F}, \mathcal{I} \cup \tilde{\mathcal{I}})$, where $\tilde{F}$ is defined as $\tilde{F}(s_t, a_t) := \text{WRAPSTEP}(F, s_t, a_t)$. This formulation does not alter the internal structure or transition dynamics of the original environment $\mathcal{E}$. Instead, it constructs an externally wrapped interface that provides the agent with a richer and more interpretable view of its operating context. The interface is denoted as $\Phi := \{\text{INFERRULES}, \text{WRAPSTEP}\}$ in the remainder of this paper.

As shown in Figure 3, the ALIGN integrates two cooperative modules, **Analyzer** and **Optimizer**, to generate aligned interfaces. The framework operates through iterative optimization, with each iteration comprising three stages: in Stage 1, the Analyzer identifies agent-environment misalignments by analyzing past interaction trajectories; in Stage 2, the Optimizer generates, validates and refines a new interface based on the detected misalignments; and in Stage 3, the agent interacts with the environment wrapped with the newly generated interface, and the failed task trajectories are fed back to Analyzer for analysis in the next iteration.

### 3.3 ALIGN FRAMEWORK

To automate the generation of interfaces that bridge the agent-environment misalignments, ALIGN need to solve two key challenges: (1) how to analyze and identify existing agent-environment misalignments, and (2) how to generate an interface that addresses these misalignments. The overall algorithm process of ALIGN is outlined in Algorithm 1 in Appendix B.

**Misalignment Analysis**  We represent each agent-environment misalignment using structured text, as shown in the bottom left of Figure 3. The "Agent High-Level Reasoning Intent" and "Environment Rule" respectively depict the agent's expectations of the action and the environment's observation rules, together representing a misalignment. The "Sufficient Observation" represents the observation the environment should provide to resolve the misalignment. To analyze and identify these misalignments, we designed the Analyzer module based on LLMs. In each iteration, the Analyzer takes the failed interaction trajectory $\tau^{(i-1)}$ in the previous iteration, the set of currently identified misalignments $\mathcal{M}$, and the interface $\Phi^{(i-1)}$ from the previous round as input, and generates a new set of misalignments $\mathcal{M}^{(i)}$. Detailed prompts for this process are provided in Appendix E.4.

**Interface Generation**  Once the new set of misalignments $\mathcal{M}^{(i)}$ is identified, we employ the Optimizer module to generate a new interface. We represent the two modules of the interface, INFERRULES and WRAPSTEP, as Python functions, as shown in the bottom right of Figure 3, to leverage the powerful code generation capabilities of LLMs. In each iteration, the Optimizer takes the newly identified misalignments $\mathcal{M}^{(i)}$ and the previous interface $\Phi^{(i-1)}$ as input, and generates a new interface $\Phi^{(i)}$. The detailed prompts for this process are provided in Appendix E.4.

**Experimental Verification**  Given the hallucination (Bang et al., 2023; Xu et al., 2024) issues of LLMs, we incorporate an experimental verification procedure. Specifically, after the Analyzer generates $\mathcal{M}^{(i)}$, it will interact with the environment wrapped by the previous interface $\Phi^{(i-1)}$ to

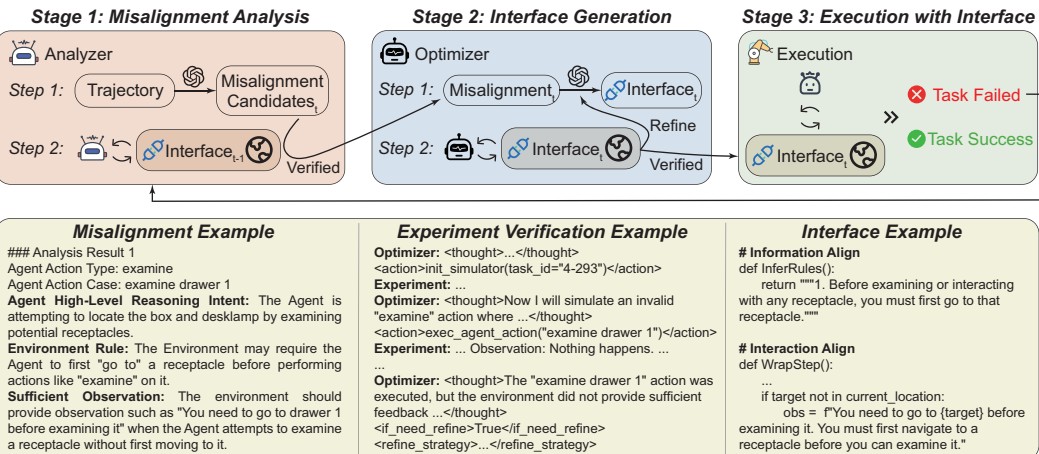

Figure 3: **ALIGN framework.** In each iteration, ALIGN progresses though three stages. **Stage 1**: the Analyzer identifies potential agent-environment misalignments and validates them through experiments; **Stage 2**: the Optimizer generates a new interface based on the previous interface and identified misalignments, followed by verification and refinement; **Stage 3**: the agent interacts with the updated interface-wrapped environment, with trajectories of failed tasks fed back to the Analyzer for analysis in the next iteration. At the bottom of the figure, examples for misalignment, verification of interface integrity by Optimizer, and the ALIGN-generated interface are provided.

validate whether the identified misalignments do indeed exist and can be resolved by the proposed "Sufficient Observation". And after the Optimizer generates the new interface $\Phi^{(i)}$, it will interact with the environment wrapped by this new interface to ensure that the generated interface can resolve the identified misalignments. If the Optimizer finds that the proposed interface is insufficient to address the discovered misalignments, it will provide a refinement strategy and regenerate the interface. This iterative process continues until the interface passes the validation, ensuring that the misalignments identified are appropriately addressed. An example of this process is provided in the bottom center of Figure 3. To facilitate this interaction with the interface-wrapped environment, we designed a set of encapsulated tools for both the Analyzer and Optimizer to use, as described in Appendix E.3.

After each iteration, the agent interacts with the environment wrapped by the new generated interface $\Phi^{(i)}$, and trajectories of the failed tasks are returned to Analyzer for further analysis. The algorithm continues iteratively until one of the following holds: (1) the pre-defined maximum number of iterations is reached; (2) no failed trajectories are produced; (3) no new misalignments are identified.

## 4 EXPERIMENT

### 4.1 EXPERIMENTAL SETTINGS

**Evaluation Protocol** To validate the effectiveness of ALIGN, we assess the performance of various agents in the original, unmodified environments. Subsequently, ALIGN is utilized to generate interfaces for these environments with the respective agents. Afterward, the agents are re-evaluated in the same environments, wrapped with the ALIGN-generated interfaces. During the interface generation and refinement process, only tasks from the training set are used. The interface logic is fixed and remains unchanged during testing. This design enables us to observe and measure the changes in agent performance before and after the interface alignment.

**Benchmarks** We conduct experiments on four representative benchmarks across three domains: embodied tasks, web navigation and tool-use. Among them, (1) ALFWorld (Shridhar et al., 2021) focuses on embodied AI agents performing household tasks through textual interactions in simulated environments; (2) ScienceWorld (Wang et al., 2022) evaluates the abilities to conduct scientific experiments and apply scientific reasoning of agents in an interactive text-based environment; (3) WebShop (Yao et al., 2022) simulates e-commerce scenarios where agents navigate product catalogs and complete purchasing tasks; and (4) M³ToolEval (Wang et al., 2024b) is specifically designed to evaluate agent performance in multi-turn tool-use tasks.

Table 1: **Effect of ALIGN-generated interfaces on four benchmarks.** For every agent we report its score without the interface (w/o ALIGN) and with the interface (w/ ALIGN); the value in parentheses is the absolute improvement. Metrics are task-success rate (%) for ALFWorld and M³ToolEval, and scores for ScienceWorld and WebShop.

| | | Embodied | | Web | Tool-use |
|---|---|---|---|---|---|
| **Method** | **Interface** | **ALFWorld** | **ScienceWorld** | **WebShop** | **M³ToolEval** |
| Vanilla | w/o ALIGN | 13.43 | 14.94 | 54.10 | 11.11 |
| | w/ ALIGN | 60.45 (+47.02) | 27.69 (+12.75) | 61.23 (+7.13) | 20.83 (+9.72) |
| ReAct | w/o ALIGN | 19.40 | 20.03 | 37.20 | 9.72 |
| | w/ ALIGN | 63.43 (+44.03) | 28.97 (+8.94) | 42.93 (+5.73) | 18.06 (+8.34) |
| Self-Consistency | w/o ALIGN | 11.94 | 14.07 | 56.23 | 11.11 |
| | w/ ALIGN | 69.40 (+57.46) | 25.41 (+11.34) | 61.10 (+4.87) | 16.67 (+5.56) |
| Self-Refine | w/o ALIGN | 3.73 | 14.87 | 44.80 | 5.55 |
| | w/ ALIGN | 40.30 (+36.57) | 22.99 (+8.12) | 52.30 (+7.50) | 6.94 (+1.39) |
| Planning | w/o ALIGN | 9.70 | 17.13 | 46.95 | 11.11 |
| | w/ ALIGN | 52.99 (+43.29) | 26.34 (+9.21) | 54.67 (+7.72) | 18.06 (+6.95) |

**Agent Methods** To verify the capability of ALIGN to enhance performance across diverse agent architectures, we evaluate five representative methods: (1) Vanilla Agent: Base implementation without specialized prompting strategies; (2) ReAct (Yao et al., 2023): Leveraging the reasoning capabilities of LLMs through interleaved reasoning and action steps; (3) Self-Consistency (Wang et al., 2023): Utilizing probabilistic outputs from LLMs to generate multiple solution paths and select the most consistent one; (4) Self-Refine (Madaan et al., 2023): Employing an iterative self-critic and refine mechanism where agents critique and refine their previous solutions; and (5) Planning Agent: Inspired by RAP Hao et al. (2023), this approach leverages the planning capabilities of LLMs to decompose complex tasks into manageable sub-tasks.

**Implementation Details** Unless otherwise noted, we use Qwen2.5-7B-Instruct (Team, 2024) as the base model of agents. The Optimizer for interface generation uses Gemini 2.5 Pro (Google, 2025), while other steps the Analyzer and Optimizer use GPT-4.1 (OpenAI, 2025). Implementation details of benchmark task splits and hyper-parameters can be found in Appendix E.

## 4.2 MAIN RESULTS

Table 1 summarizes the task success rates or scores of five representative agent methods in the environment without (w/o) or with (w/) ALIGN-generated interface. The interfaces generated and the misalignments analyzed can be found in Appendix F and the token consumption analysis can be found in Appendix D. Our empirical investigation yields three principal findings:

**(1) ALIGN consistently enhances performance across different domains.** All evaluated agent methods demonstrate significant performance improvements when utilizing ALIGN-generated interfaces. Specifically, the five agent methods exhibit mean improvements of 45.67% in task-success rate for ALFWorld, 10.07 points for ScienceWorld, 6.59 points for WebShop, and 6.39% in task-success rate for M³ToolEval. These consistent improvements substantiate the effectiveness of ALIGN.

**(2) Agent-environment misalignment is a pervasive phenomenon impeding the agent performance.** The observed performance enhancements provide empirical evidence that numerous errors in baseline configurations originate from implicit constraints or under-specified observation, rather than from intrinsic reasoning deficiencies. This finding suggests that when these environmental constraints are explicitly surfaced, agents can execute their intended tasks with substantially improved reliability. Consequently, we posit that agent-environment misalignment is pervasive in interactive decision-making tasks, and addressing this problem is crucial for advancing agent performance.

**(3) Alignment between agent and environment can facilitate identification of additional performance-influencing factors.** While the Self-Consistency agent achieves a 69.40% success rate in ALFWorld with ALIGN, the performance of Self-Refine agent remains comparatively sub-optimal (40.30%), indicating potential deficiencies in the critic and self-refinement capabilities of the Qwen2.5-7B-Instruct model. These limitations are similarly manifested in the M³ToolEval results. Furthermore, the relatively modest performance improvements in ScienceWorld suggest that Qwen2.5-7B-Instruct may exhibit insufficient scientific causal reasoning capabilities. These

observations indicate that properly aligning agent and environment enables more precise isolation and analysis of other factors influencing agent performance beyond alignment considerations.

## 4.3 INTERFACE QUALITY ANALYSIS

Table 2: **Impact of the ALIGN-generated interface on consecutive invalid actions.** The metric reports the fraction (%) of consecutive invalid actions. Lower values indicate more desirable behavior. $\Delta$ denotes the relative reduction with respect to the **w/o ALIGN** setting.

| Method | ALFWorld | | | ScienceWorld | | |
|---|---|---|---|---|---|---|
| | **w/o ALIGN** | **w/ ALIGN** | $\Delta$ | **w/o ALIGN** | **w/ ALIGN** | $\Delta$ |
| Vanilla | 77.91 | 26.59 | 66% | 49.12 | 24.47 | 50% |
| ReAct | 82.23 | 38.63 | 53% | 46.61 | 29.99 | 36% |
| Self-Consistency | 77.71 | 15.08 | 81% | 51.10 | 31.51 | 38% |
| Self-Refine | 90.38 | 45.84 | 49% | 58.02 | 29.48 | 49% |
| Planning | 74.09 | 19.14 | 74% | 68.67 | 20.94 | 70% |
| **Average** | 80.46 | 28.51 | 65% | 54.70 | 27.28 | 49% |

**Influence on Agent Decision** To quantitatively assess the influence of ALIGN-generated interfaces on agent decision beyond end-task performance metric, we introduce a metric that measures the frequency of *consecutive invalid actions* by calculating the proportion of the actions that occur within sequences of two or more consecutive `invalid` steps. Lower values of this metric indicate: (1) enhanced agent awareness of implicit preconditions, and (2) improved recovery capability following isolated errors. Table 2 presents the results for five agent methods implemented on ALFWorld and ScienceWorld. The empirical results demonstrate a substantial reduction in consecutive invalid actions frequency across all agent methods when utilizing ALIGN-generated interfaces. Specifically, we observe a mean reduction of 65% in ALFWorld and 49% in ScienceWorld. These findings provide robust evidence that ALIGN effectively enriches the information conveyed by the observation, preventing agents from entering repetitive error cycles, which aligns with the findings documented in Section 4.2.

**Comparison with Agentic Systems and Human-designed Interfaces** To further assess the effectiveness of our automatically generated interfaces, we compare ALIGN against (1) agentic frameworks equipped with carefully designed reasoning, planning and memory modules and (2) human-designed interfaces. The experimental setup and results are presented in Appendix C.2. As shown in Table 7, even without bespoke reasoning, planning, or memory modules, a vanilla agent that directly outputs the next action yields a 6.71 percentage points higher success rate than the best agentic system when paired with ALIGN-generated interfaces, indicating agent-environment misalignment substantially constrains the performance of LLM-based agents in interactive tasks. Moreover, using interfaces automatically generated by ALIGN yields a 13.44 percentage points higher success rate than human-designed interfaces, further validating the effectiveness of our method (Table 8).

## 4.4 GENERALIZATION AND GENERALITY STUDY

**Generalization Study** To evaluate the generalization capabilities of ALIGN, we performed the following two experiments, with the results presented in Table 3 and detailed results in Appendix C.3.

(1) ALIGN can generalize to different agent architectures. Panel (a) of Table 3 applies interfaces generated with the Vanilla agent to the other four agents. Across all four environments every target agent shows consistent growth, demonstrating that ALIGN captures genuine and previously unexposed environment constraints. This also reinforces the earlier conclusion that agent-environment misalignment is a pervasive source of error independent of the agent's reasoning style.

(2) ALIGN can generalize to larger and heterogeneous LLMs. Panel (b) of Table 3 examines whether an interface generated with Qwen2.5-7B-Instruct can extend to larger or architecturally different model backbones. The results demonstrate that ALIGN-generated interfaces lead to performance improvements across base models of varying sizes and architectural families, which indicates that our method possesses strong generalization capabilities. We also observe that this generalization is not uniformly robust across all model families and datasets. For instance, Llama3.1-8B-Instruct (Meta, 2025a) shows only a marginal gain of +0.33 on the WebShop benchmark. This limited improvement may be attributed to the inherent reasoning capabilities of the model itself.

Table 3: **Generalization of ALIGN-generated interfaces across agents and models.** Mean performance improvements from applying ALIGN-generated interfaces in the four environments across different settings. (a) Cross-agent transfer: interfaces generated with a Vanilla agent improve other agent methods. (b) Cross-model transfer: interfaces generated with Qwen2.5-7B-Instruct can generalize to other LLMs.

| (a) Interface source: Vanilla agent | | | | |
|---|---|---|---|---|
| Target method | ALF. | Sci. | Web. | M³T. |
| ReAct | +39.56 | +12.29 | +7.87 | +5.56 |
| Self-Consistency | +51.49 | +15.30 | +3.00 | +8.33 |
| Self-Refine | +34.33 | +14.11 | +6.17 | +4.17 |
| Planning | +41.05 | +9.66 | +3.26 | +11.11 |
| (b) Interface source: Qwen2.5-7B-Instruct agent | | | | |
| Target LLM | ALF. | Sci. | Web. | M³T. |
| Qwen2.5-14B-Instruct | +17.46 | +4.61 | +4.66 | +6.11 |
| Llama3.1-8B-Instruct | +5.97 | +10.27 | +0.33 | +0.83 |
| Llama3.3-70B-Instruct | +5.82 | +3.99 | +5.68 | +1.67 |

Table 4: **Generality of ALIGN.** Task success rates (SR) without and with ALIGN-generated interfaces in ALFWorld across two settings. (a) Using GPT-4.1 series models as the base model of agents; (b) Using GiGPO-Qwen2.5-7B-Instruct evaluated under different agent architectures.

| (a) GPT-4.1 series | | |
|---|---|---|
| Base Model | Interface | SR (%) |
| GPT-4.1-mini | w/o ALIGN | 28.36 |
| | w/ ALIGN | 64.93 (+36.57) |
| GPT-4.1 | w/o ALIGN | 73.88 |
| | w/ ALIGN | 93.28 (+19.40) |
| (b) GiGPO-Qwen2.5-7B-Instruct | | |
| Agent Method | Interface | SR (%) |
| Vanilla | w/o ALIGN | 35.04 |
| | w/ ALIGN | 55.97 (+20.93) |
| Training Config | w/o ALIGN | 89.55 |
| | w/ ALIGN | 92.54 (+2.99) |

Taken together, these results show that ALIGN-generated interfaces can generalize (1) across agent policies and (2) across model scales and families, validating the practicality of ALIGN.

**Generality Study** In this work, our empirical observations indicate that the root cause of agent-environment misalignment lies in the robustness of the interface itself, making it a universal issue that affects agents irrespective of the underlying model capability. To further validate this claim and assess the generality of ALIGN, we conduct experiments on both closed-source LLMs and domain-specific models trained within the environment. For the former, we use the GPT-4.1 series; for the latter, we use GiGPO-Qwen2.5-7B-Instruct-ALFWorld (Feng et al., 2025), a state-of-the-art model specifically post-trained on ALFWorld via reinforcement learning. Detailed experimental setup and full results are provided in Appendix C.4. As the results reported in Panel (a) of Table 4 shown, applying the ALIGN-generated interface substantially improves the performance of the GPT-4.1-based agent from 73.88% to 93.28%. Meanwhile, as the results reported in Panel (b) of Table 4 shown, the ALIGN-generated interface also enhances the performance of the domain-specific model under both our Vanilla Agent setting and its original training configuration, from 35.04% to 55.97% and 89.55% to 92.54%, respectively. These findings demonstrate that the fundamental and pervasive nature of agent-environment misalignment stems from deficiencies in the environment's interface rather than solely from the reasoning limitations of any given model, and further corroborate the generality of our method across both frontier and domain-specialized models.

## 4.5 ABLATION STUDY

**Ablation on Interface Components** Starting from the full ALIGN interface, we conduct two ablations: (1) w/o INFERRULES and (2) w/o WRAPSTEP. Table 5 reports the change relative to the full interface on ALFWorld and ScienceWorld, with the full results presented in Appendix C.5. Both ablations reduce performance: w/o INFERRULES averages -6.72 percentage points on ALFWorld and -2.05 on ScienceWorld, while removing WRAPSTEP yields a larger decline of -31.79 percentage points and -7.84, respectively. These decreases confirm that each interface component contributes meaningfully. Moreover, the

Table 5: **Ablation on Interface components.** Values represent the change in success rate (%) on ALFWorld and score on ScienceWorld. Negative values mean performance drops from the *Full* interface.

| | w/o INFERRULES | | w/o WRAPSTEP | |
|---|---|---|---|---|
| Method | ALF. | Sci. | ALF. | Sci. |
| Vanilla | -8.96 | -3.35 | -33.58 | -4.72 |
| ReAct | -5.22 | -2.08 | -17.91 | -6.44 |
| Self-Consistency | -1.49 | -2.30 | -37.27 | -10.59 |
| Self-Refine | -7.46 | -1.72 | -34.33 | -7.59 |
| Planning | -10.45 | -0.78 | -26.87 | -9.86 |
| *Mean* | -6.72 | -2.05 | -31.79 | -7.84 |

much larger drop w/o WRAPSTEP shows the critical role of fine-grained and enriched observation during interaction. This also suggests that rich, LLM-friendly observation should be prioritized by future environment designers when constructing environments.

**Ablation on Experimental Verification** To assess whether the experimental verification procedure in Section 3.3 is indispensable, we ablated it and re-ran ALIGN with the Vanilla agent on ALFWorld. As a surrogate, we employed a multi-sampling strategy in each iteration: the Analyzer sampled six candidate misalignments and selected the one it judged most accurate; the Optimizer then sampled six candidate interfaces and likewise chose its top candidate. Within this multi-sampling process, we controlled stochasticity via decoding temperature; specifically, we evaluated $T \in \{0.2, 0.5\}$ under the prompts listed in Appendix E.4. The resulting task success rates over three iterations are summarized in Table 6. Without the ability to execute experiments, task success rate deteriorates sharply, a result of the limited single-shot reliability of LLMs in both diagnosing misalignments and synthesizing correct interfaces, which underscores the necessity of the experimental verification procedure design.

Table 6: Task success rate (%) on ALFWorld across iterations without experimental verification procedure.

| Temp. | Iter0 | Iter1 | Iter2 | Iter3 |
|---|---|---|---|---|
| 0.2 | 13.43 | 22.39 | 0.00 | 0.00 |
| 0.5 | 13.43 | 23.88 | 1.49 | 0.75 |

## 5 CONCLUSION

In this work, we introduce **ALIGN**, a novel framework that automatically generates aligned interfaces to alleviate the **agent-environment misalignment**, a pervasive and underexplored source of failure in interactive decision-making tasks. By diagnosing implicit constraints through the Analyzer and synthesizing aligned interface via the Optimizer, ALIGN improves agent performance significantly on four representative benchmarks across three domains: embodied tasks, web navigation, and tool-use. Our results demonstrate that ALIGN not only boosts performance across multiple agent methods but also generalizes effectively to unseen models and strategies, offering a robust, plug-and-play solution that decouples agent designs from manual environment-specific alignment. These findings suggest that automatic interface generation is a promising direction for building more reliable, reusable, and interpretable LLM-based agents. Future research should explore richer forms of interface representation, expand evaluations to more domains, and develop finer-grained metrics to quantify interface quality and its impact on agent behavior.

### LIMITATIONS AND FUTURE WORK

Despite the effectiveness of ALIGN in alleviating agent-environment misalignment, this work represents an initial exploration into automated interface generation. Several important directions remain open for further investigation:

**Toward more diverse and complex environments.** Our current evaluation focuses on environments with discrete, text-based action spaces across three domains: embodied tasks, web navigation, and tool-use. ALIGN's applicability to more complex settings remains to be explored. Future work could investigate more complex environments like extending ALIGN to multimodal domains such as GUI agents, where interfaces must process visual observations alongside textual feedback.

**Beyond information and observation augmentation.** As formalized in Section 3.1, a complete interface comprises three components: $f_{\text{info}}$, $f_{\text{obs}}$, and $f_{\text{act}}$. This work focuses on optimizing $f_{\text{info}}$ and $f_{\text{obs}}$ to alleviate the agent-environment misalignment. However, $f_{\text{act}}$ also plays a critical role in interactive tasks. Constraining agents to predefined action spaces may force them to deviate from their natural output distributions, potentially degrading performance. Automatically generating and optimizing $f_{\text{act}}$ to bridge the gap between an agent's preferred action representation and the environment's expected format remains an important direction.

**Metrics for interface quality.** This paper evaluates interface effectiveness primarily through task success rates and consecutive invalid actions. More comprehensive metrics are needed to quantify interface influence on agent behavior. Promising directions include: (1) developing finer-grained behavioral diagnostics measuring specific aspects of agent understanding, such as exploratory actions or strategy diversity; (2) employing LLM-as-a-Judge (Zheng et al., 2023) paradigms to evaluate whether interfaces successfully convey environment constraints.

## REPRODUCIBILITY STATEMENT

We present the framework and algorithm design of our method in Section 3 and Appendix B, and the implementation details of the experiments in Appendix C and Appendix E. Meanwhile, the code necessary to reproduce the proposed methods and the main experiments has been provided as supplemental material. The supplemental material also includes the corresponding experimental logs.

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

## A    LLM USAGE STATEMENT

Throughout the completion of this work, the LLM was employed solely for the purpose of refining sentences and improving grammatical accuracy during the manuscript writing process.

## B    FORMALIZATION OF THE ALIGN ALGORITHM

The formalization of the ALIGN algorithm is outlined in Algorithm 1.

---

**Algorithm 1** ALIGN: Auto-Aligned Interface Generation

---

**Require:** Environment $\mathcal{E}$, Agent $\pi$, Task training set $\mathcal{T}_{\text{train}}$, Maximum iterations $K$
1: Initialize misalignment set $\mathcal{M} \leftarrow \emptyset$, interface $\Phi^{(0)} \leftarrow \{\text{INFERRULES}^{(0)}, \text{WRAPSTEP}^{(0)}\}$, where $\text{INFERRULES}^{(0)}$ and $\text{WRAPSTEP}^{(0)}$ are identity functions
2: **for** $i = 1, 2, \ldots, K$ **do**
3:     $\tilde{\mathcal{E}}^{(i-1)} \leftarrow$ Environment $\mathcal{E}$ wrapped with interface $\Phi^{(i-1)}$
4:     $\tau_{\text{fail}}^{(i-1)} \leftarrow$ Failed trajectories from agent $\pi$ interacting with $\tilde{\mathcal{E}}^{(i-1)}$ on $\mathcal{T}_{\text{train}}$
5:     **if** $\tau_{\text{fail}}^{(i-1)} = \emptyset$ **then**
6:         **break**                                      ▷ No more failures in the training set
7:     **end if**
        // Stage 1: Misalignment Analysis
8:     $\mathcal{M}^{(i)} \leftarrow$ Analyzer$(\tau_{\text{fail}}^{(i-1)}, \mathcal{M}, \Phi^{(i-1)})$
9:     **if** $\mathcal{M}^{(i)} = \emptyset$ **then**
10:         **break**                                    ▷ No new misalignments identified
11:     **end if**
12:     $\mathcal{M} \leftarrow \mathcal{M} \cup \mathcal{M}^{(i)}$
        // Stage 2: Interface Generation
13:     $\Phi^{(i)} \leftarrow$ Optimizer$(\mathcal{M}^{(i)}, \Phi^{(i-1)})$
14: **end for**
15: **return** final interface $\Phi^{(i)}$

---

## C    SUPPLEMENTARY EXPERIMENTAL SETUP AND DETAILED RESULTS

### C.1    PRELIMINARY EXPERIMENTS

To preliminarily assess the significance of agent-environment misalignment, we conducted exploratory experiments on the ALFWorld. We employed the vanilla Qwen2.5-7B-Instruct agent with a temperature setting of 0.0. The deployment protocol, prompt template, followed the same configuration described in Appendix E and Appendix E.4.

During the experiments, we introduced a minor modification to the environment: if the agent issued the action *examine receptacle* and the environment returned the default observation "Nothing happens.", we replaced it with "You need to first go to receptacle before you can examine it." This simple adjustment increased the agent's task success rate from 13.4% to 31.3%.

### C.2    INTERFACE QUALITY ANALYSIS EXPERIMENTS

To further assess the quality of the ALIGN-generated interface, we first compare our method with human-designed agentic system. Our experiments are conducted on ALFWorld using the AgentSquare (Shang et al., 2025) framework. To maximize the advantages of the agentic system, we adopt gpt-4.1-2025-04-14 as the base model, select OPENAGI (Ge et al., 2023) for the planning

Table 7: Experimental results of the comparison between agents with ALIGN-generated interface and agents with human-designed reasoning, planning and memory module.

| Agent Framework | Interface | Memory Module | pick and place | pick clean and place | pick heat and place | pick cool and place | look at / examine in light | pick two obj and place | Success Rate (%) |
|---|---|---|---|---|---|---|---|---|---|
| AgentSquare | / | Generative | 95.83 | 87.10 | 69.57 | 95.24 | 83.33 | 88.24 | 86.57 |
| AgentSquare | / | DiLu | 91.67 | 87.10 | 52.17 | 95.24 | 83.33 | 70.59 | 80.60 |
| AgentSquare | / | TP | 87.50 | 51.61 | 4.35 | 61.90 | 27.78 | 47.06 | 47.76 |
| AgentSquare | / | VOYAGER | 95.83 | 83.87 | 52.17 | 90.48 | 83.33 | 64.71 | 79.10 |
| Vanilla Agent | w/o ALIGN | / | 100.00 | 93.55 | 13.04 | 71.43 | 61.11 | 100.00 | 73.88 |
| Vanilla Agent | w/ ALIGN | / | 100.00 | 100.00 | 78.26 | 100.00 | 77.78 | 100.00 | **93.28** |

module, Self-Refine (Madaan et al., 2023) for the reasoning module, and evaluate memory using Generative (Park et al., 2023), DiLu (Wen et al., 2024), TP (Yu et al., 2024), and VOYAGER (Wang et al., 2024a). For our approach, we employ a gpt-4.1-2025-04-14-based vanilla agent, where the interface is generated with the gpt-4.1-2025-04-14-mini-based vanilla agent by ALIGN (the experimental setup is same as Appendix C.4). The results are reported in Table 7.

Furthermore, we compare the ALIGN-generated interface against the human-designed interface. We adopt the following configurations for comparison with our method: (1) **Few-shot:** Settings identical to those in the ReAct (Yao et al., 2023); (2) **Valid Actions:** Supplying the agent with all valid actions at every response turn, analogous to the `check_valid_actions` configuration in Agent-Board (Ma et al., 2024); (3) **Human-Designed Interface:** Interfaces manually crafted by Ph.D. students after inspecting ALFWorld experiments, examining trajectories, and running experiments themselves. The design logic includes: executing "go to" prior to each action; automatically checking object labels; converting "put" to "move" when appropriate;

Table 8: Experimental results of the comparison between agents with ALIGN-generated interface and agents with human-designed interfaces.

| Experimental Setting | Success Rate (%) |
|---|---|
| w/o Interface | 13.43 |
| Few-shot | 44.78 |
| Valid Actions | 44.03 |
| Human Designed Interface | 47.01 |
| ALIGN-generated Interface | **60.45** |

returning the action space upon invalid actions; issuing reminders when "clean with" is applied to non-sinkbasin objects; and other hand-engineered rules. We use Qwen2.5-7B-Instruct as the base model. Experimental results are reported in Table 8.

## C.3 GENERALIZATION STUDY EXPERIMENTS

Detailed results of the generalization study are provided for the cross-method experiments in Table 9 and for the cross-model experiments in Tables 10, 11, and 12.

Table 9: **Generalization of ALIGN-generated interfaces generated with Vanilla agents to other agent methods.** For each agent we report its score without the interface (w/o ALIGN) and with the interface (w/ ALIGN); the value in parentheses is the *absolute* improvement.

| Base Method: Vanilla | | Embodied | | Web | Tool-use |
|---|---|---|---|---|---|
| Method | Interface | ALFWorld | ScienceWorld | WebShop | M³ToolEval |
| ReAct | w/o ALIGN | 19.40 | 20.03 | 37.20 | 9.72 |
| | w/ ALIGN | 58.96 (+39.56) | 32.32 (+12.29) | 45.07 (+7.87) | 15.28 (+5.56) |
| Self-Consistency | w/o ALIGN | 11.94 | 14.07 | 56.23 | 11.11 |
| | w/ ALIGN | 63.43 (+51.49) | 29.37 (+15.30) | 59.23 (+3.00) | 19.44 (+8.33) |
| Self-Refine | w/o ALIGN | 3.73 | 14.87 | 44.80 | 5.55 |
| | w/ ALIGN | 38.06 (+34.33) | 28.98 (+14.11) | 50.97 (+6.17) | 9.72 (+4.17) |
| Planning | w/o ALIGN | 9.70 | 17.13 | 46.95 | 11.11 |
| | w/ ALIGN | 50.75 (+41.05) | 26.79 (+9.66) | 50.21 (+3.26) | 22.22 (+11.11) |

Table 10: **Generalization of ALIGN-generated interfaces generated with Qwen2.5-7B-Instruct to Qwen2.5-14B-Instruct.** For each agent we report its score without the interface (w/o ALIGN) and with the interface (w/ ALIGN); the value in parentheses is the *absolute* improvement.

| Base Model: Qwen2.5-14B-Instruct | | Embodied | | Web | Tool-use |
|---|---|---|---|---|---|
| Method | Interface | ALFWorld | ScienceWorld | WebShop | M³ToolEval |
| Vanilla | w/o ALIGN | 48.51 | 22.58 | 53.67 | 13.89 |
| | w/ ALIGN | 52.24 (+3.73) | 37.58 (+15.00) | 58.40 (+4.73) | 18.06 (+4.17) |
| ReAct | w/o ALIGN | 54.48 | 31.24 | 39.73 | 15.28 |
| | w/ ALIGN | 70.15 (+15.67) | 29.79 (-1.45) | 42.17 (+2.44) | 26.39 (+11.11) |
| Self-Consistency | w/o ALIGN | 43.28 | 25.60 | 52.63 | 13.89 |
| | w/ ALIGN | 72.39 (+29.11) | 26.68 (+1.08) | 51.07 (-1.56) | 27.78 (+13.89) |
| Self-Refine | w/o ALIGN | 5.22 | 18.97 | 41.00 | 15.28 |
| | w/ ALIGN | 14.18 (+8.96) | 20.72 (+1.75) | 39.93 (-1.07) | 16.67 (+1.39) |
| Planning | w/o ALIGN | 49.25 | 21.46 | 31.72 | 25.00 |
| | w/ ALIGN | 79.10 (+29.85) | 28.13 (+6.67) | 50.47 (+18.75) | 25.00 (0.00) |

Table 11: **Generalization of ALIGN-generated interfaces generated with Qwen2.5-7B-Instruct to Llama3.1-8B-Instruct.** For each agent we report its score without the interface (w/o ALIGN) and with the interface (w/ ALIGN); the value in parentheses is the *absolute* improvement.

| Base Model: Llama3.1-8B-Instruct | | Embodied | | Web | Tool-use |
|---|---|---|---|---|---|
| Method | Interface | ALFWorld | ScienceWorld | WebShop | M³ToolEval |
| Vanilla | w/o ALIGN | 5.22 | 23.59 | 35.17 | 5.56 |
| | w/ ALIGN | 14.18 (+8.96) | 36.40 (+12.81) | 24.00 (-11.17) | 1.39 (-4.17) |
| ReAct | w/o ALIGN | 1.49 | 22.42 | 27.12 | 12.50 |
| | w/ ALIGN | 15.67 (+14.18) | 28.74 (+6.32) | 27.10 (-0.02) | 22.22 (+9.72) |
| Self-Consistency | w/o ALIGN | 5.22 | 25.21 | 29.80 | 4.17 |
| | w/ ALIGN | 11.94 (+6.72) | 34.83 (+9.62) | 15.83 (-13.97) | 2.78 (-1.39) |
| Self-Refine | w/o ALIGN | 0.00 | 22.34 | 27.70 | 1.39 |
| | w/ ALIGN | 0.75 (+0.75) | 31.33 (+8.99) | 37.43 (+9.73) | 1.39 (0.00) |
| Planning | w/o ALIGN | 6.72 | 13.33 | 23.67 | 4.17 |
| | w/ ALIGN | 5.97 (-0.75) | 26.95 (+13.62) | 40.77 (+17.10) | 4.17 (0.00) |

Table 12: **Generalization of ALIGN-generated interfaces generated with Qwen2.5-7B-Instruct to Llama3.3-70B-Instruct.** For each agent we report its score without the interface (w/o ALIGN) and with the interface (w/ ALIGN); the value in parentheses is the *absolute* improvement.

| Base Model: Llama3.3-70B-Instruct | | Embodied | | Web | Tool-use |
|---|---|---|---|---|---|
| Method | Interface | ALFWorld | ScienceWorld | WebShop | M³ToolEval |
| Vanilla | w/o ALIGN | 52.99 | 55.77 | 51.67 | 37.50 |
| | w/ ALIGN | 43.28 (-9.71) | 57.74 (+1.97) | 62.07 (+10.40) | 33.33 (-4.17) |
| ReAct | w/o ALIGN | 45.52 | 56.50 | 58.22 | 34.72 |
| | w/ ALIGN | 47.01 (+1.49) | 58.28 (+1.78) | 53.83 (-4.39) | 43.06 (+8.34) |
| Self-Consistency | w/o ALIGN | 54.48 | 56.66 | 50.37 | 36.11 |
| | w/ ALIGN | 65.67 (+11.19) | 59.24 (+2.58) | 55.63 (+5.26) | 34.72 (-1.39) |
| Self-Refine | w/o ALIGN | 38.06 | 56.97 | 38.40 | 1.39 |
| | w/ ALIGN | 46.27 (+8.21) | 60.17 (+3.20) | 47.85 (+9.45) | 0.00 (-1.39) |
| Planning | w/o ALIGN | 58.96 | 48.75 | 54.90 | 33.33 |
| | w/ ALIGN | 76.87 (+17.91) | 59.17 (+10.42) | 62.60 (+7.70) | 40.28 (+6.95) |

## C.4 GENERALITY STUDY EXPERIMENTS

Table 13: Experimental results for GPT-4.1 series agents with ALIGN on ALFWorld.

| Base Model | Interface | pick and place | pick clean and place | pick heat and place | pick cool and place | look at / examine in light | pick two obj and place | Success Rate (%) |
|---|---|---|---|---|---|---|---|---|
| gpt-4.1-mini | w/o ALIGN | 58.33 | 22.58 | 8.70 | 9.52 | 22.22 | 52.94 | 28.36 |
| | w/ ALIGN | 95.83 | 87.10 | 26.09 | 80.95 | 27.78 | 52.94 | **64.93** |
| gpt-4.1 | w/o ALIGN | 100.00 | 93.55 | 13.04 | 71.43 | 61.11 | 100.00 | 73.88 |
| | w/ ALIGN | 100.00 | 100.00 | 78.26 | 100.00 | 77.78 | 100.00 | **93.28** |

For the validation on closed-source LLMs, we selected the GPT-4.1 family. Specifically, we experimented with gpt-4.1-mini-2025-04-14 and gpt-4.1-2025-04-14. First, we used gpt-4.1-mini-2025-04-14 as the base model to instantiate a Vanilla Agent and synthesize interface with ALIGN. We then applied the same interface to an agent powered by gpt-4.1-2025-04-14. All other experimental settings were identical to those in the main experiments. The results are presented in Table 13.

For domain-specific models trained within the environment, we used GiGPO-Qwen2.5-7B-Instruct-ALFWorld, a state-of-the-art model post-trained on ALFWorld via reinforcement learning (Feng et al., 2025). We reused the interface produced in our main experiment (generated with the base Qwen2.5-7B-Instruct model under the Vanilla Agent method). At evaluation time, we considered two configurations: (1) our Vanilla Agent setting, and (2) a configuration that matches the logic and prompt setting used during training in the original paper.

## C.5 ABLATION STUDY EXPERIMENTS

The full result of interface ablation experiment can be found in Table 14.

Table 14: Ablation study on the components of ALIGN. Values represent task success rates (%) or scores. For ablated conditions (w/o INFERRULES, w/o WRAPSTEP), performance changes from the 'Full' are shown in parentheses.

| Method | Interface | Embodied | | Web | Tool |
|---|---|---|---|---|---|
| | | **ALFWorld** | **ScienceWorld** | **Webshop** | **M³ToolEval** |
| Vanilla | Full | 60.45 | 27.69 | 61.23 | 20.83 |
| | w/o INFERRULES | 51.49 (-8.96) | 24.34 (-3.35) | 51.03 (-10.20) | 18.06 (-2.77) |
| | w/o WRAPSTEP | 26.87 (-33.58) | 22.97 (-4.72) | 61.23 (-0.00) | 11.11 (-9.72) |
| ReAct | Full | 63.43 | 28.97 | 42.93 | 18.06 |
| | w/o INFERRULES | 58.21 (-5.22) | 26.89 (-2.08) | 35.97 (-6.96) | 9.72 (-8.34) |
| | w/o WRAPSTEP | 45.52 (-17.91) | 22.53 (-6.44) | 47.60 (+4.67) | 19.44 (+1.38) |
| Self-Consistency | Full | 69.40 | 25.41 | 61.10 | 16.67 |
| | w/o INFERRULES | 67.91 (-1.49) | 23.11 (-2.30) | 55.67 (-5.43) | 13.89 (-2.78) |
| | w/o WRAPSTEP | 23.13 (-17.91) | 14.82 (-10.59) | 60.67 (-0.43) | 15.28 (-1.39) |
| Self-Refine | Full | 40.30 | 22.99 | 52.30 | 6.94 |
| | w/o INFERRULES | 32.84 (-7.46) | 21.27 (-1.72) | 46.33 (-5.97) | 6.94 (-0.00) |
| | w/o WRAPSTEP | 5.97 (-34.33) | 15.40 (-7.59) | 47.80 (-4.50) | 6.94 (-0.00) |
| Planning | Full | 52.99 | 26.34 | 54.67 | 18.06 |
| | w/o INFERRULES | 42.54 (-10.45) | 25.56 (-0.78) | 48.18 (-6.49) | 16.67 (-1.39) |
| | w/o WRAPSTEP | 26.12 (-26.87) | 16.48 (-9.86) | 52.87 (-1.80) | 16.67 (-1.39) |

# D TOKEN CONSUMPTION ANALYSIS

The average token consumption per iteration in the main experiment described in Section 4.1 is shown in Table 15.

Due to the "Experimental Verification" setup, the Analyzer and Optimizer need to interact with the environment multiple times, and all previous interaction histories are included as new prompt inputs to

Table 15: The average token consumption per iteration in the main experiment described in Section 4.1.

|  |  | **ALFWorld** | **ScienceWorld** | **WebShop** | **M³ToolEval** |
|---|---|---|---|---|---|
| **Analyzer** | Input Token (M) | 0.2770 | 0.4333 | 0.1783 | 0.1094 |
|  | Output Token (M) | 0.0040 | 0.0036 | 0.0048 | 0.0016 |
|  | Total Token (M) | 0.2809 | 0.4370 | 0.1831 | 0.1109 |
| **Optimizer** | Input Token (M) | 0.2619 | 0.2288 | 0.0669 | 0.1100 |
|  | Output Token (M) | 0.0087 | 0.0172 | 0.0040 | 0.0118 |
|  | Total Token (M) | 0.2706 | 0.2460 | 0.0709 | 0.1217 |
| **Total** | Total Token (M) | 0.5515 | 0.6830 | 0.2540 | 0.2326 |

the LLM in each round of interaction. Additionally, when the Optimizer identifies that the generated interface is imperfect, it needs to refine the previously generated interface and conduct experimental verification again, leading to increased token consumption. However, as LLM capabilities continue to improve and hallucination issues decrease, this cost will gradually reduce. Furthermore, it is worth noting that:

- The INFERRULES wrapper and WRAPSTEP wrapper are implemented as python logic code, which does not involve calls to models or agents, therefore not incurring additional token consumption. On the contrary, as demonstrated in our experiments in Section 4.3, **using ALIGN-generated interfaces can help agents reduce repetitive meaningless actions, thereby reducing the number of LLM calls and decreasing token consumption** compared to not using ALIGN-generated interfaces.
- Except when the Optimizer generates interface codes requiring the cutting edge LLMs (such as Gemini 2.5 Pro), weaker and more cost-effective LLMs (such as GPT-4.1-mini) can be used at other times, which will significantly reduce the operational costs of ALIGN.
- ALIGN-generated interfaces can generalize to different agent architectures and base LLMs. This means that for each environment, using the ALIGN method to generate an interface only once can bring performance improvements to different agents, regardless of agent version updates. This also means that **the cost of interface generation is a one-time expense**, rather than requiring the generation of new interfaces for each task execution. Therefore, from an amortization perspective, the method's cost becomes increasingly economical as the environment is utilized more frequently, with the one-time interface design cost being distributed across multiple uses and becoming proportionally smaller with increased usage.

# E  IMPLEMENTATION DETAILS

## E.1  BENCHMARKS TASK SPLITS

The task splits of benchmarks we use are as follows:

(1) ALFWorld (Shridhar et al., 2021): We adhere to the original dataset partitioning presented in the paper, wherein the tasks from the "eval_out_of_distribution" category are used as the test set, and the "train" category is designated as the training set. In each iteration, we randomly select three tasks from the training set of each task type to serve as the training data for the agent's interaction.

(2) ScienceWorld (Wang et al., 2022):We follow the original partitioning of the train and test sets as described in the paper. For efficiency reasons, during testing, we select at most the first five tasks from the 30 available task types for evaluation. In each iteration, we randomly select one task from the training set of each task type to be used as the training data for the agent's interaction.

(3) WebShop (Yao et al., 2022): In alignment with the setup of Yao et al. (2023), we use tasks with IDs ranging from 0 to 49 (50 tasks in total) as the test set, and tasks with IDs from 50 to 199 (150 tasks in total) as the training set. In each iteration, we randomly select 20 tasks from the training set to serve as the training data for the agent's interaction.

(4) M³ToolEval (Wang et al., 2024b): Since M³ToolEval does not provide a distinct training set division, we select two tasks from each task type in the original dataset as the training set, with the remaining tasks used as the test set. In each iteration, the entire training set is utilized for the agent's interaction.

### E.2 HYPERPARAMETER AND EXPERIMENT SETTING

For all the agents, we deploy them uniformly using vllm (Kwon et al., 2023) across 8 Nvidia A100 80GB GPUs, with the inference temperature set to 0.0. The models utilized contain Qwen2.5-7B-Instruct[2] (Team, 2024), Qwen2.5-14B-Instruct[3] (Team, 2024), Llama3.1-8B-Instruct[4] (Meta, 2025a) and Llama3.3-70B-Instruct[5] (Meta, 2025b).

In ALIGN, we use Gemini 2.5 Pro (gemini-2.5-pro-exp-03-25)(Google, 2025) for Optimizer to generate new interface, with the temperature set to 0.2. For other scenarios requiring the use of an LLM, we employ GPT-4.1 (gpt-4.1-2025-04-14)(OpenAI, 2025). We set $K = 8$ during experiments.

### E.3 TOOLS FOR EXPERIMENTAL VERIFICATION

In order to implement the experimental verification process mentioned in Section 3.3, we have encapsulated the following tools for Analyzer and Optimizer to interact with the interface-wrapped environment:

(1) `init_simulator(task_id, interface)`: Initializes an experimental task, specifying the task ID and the interface code.

(2) `reset_simulator()`: Resets the experimental task.

(3) `run_task()`: Runs the current task until completion, returning the interaction trajectory.

(4) `exec_agent_action(agent_action)`: Executes a specific action and returns the enhanced observation after the interface processing.

(5) `get_agent_action()`: Based on the current trajectory, returns the next action to be issued by the agent.

(6) `change_obs(obs)`: Modifies the observation of the previous action execution.

### E.4 PROMPT TEMPLATES

We present the prompt template of the Analyzer and Optimizer for ALFWorld. For the prompt templates of other benchmarks, please refer to the supplemental materials. For the WebShop and M³ToolEval environments, no "Gold Action and Observation Sequence" is provided.

---

**Analyzer Prompt Template of Misalignment Analysis**

**User message:**
In modern benchmarks evaluating LLM Agent reasoning capabilities, human designers create an Environment with a set of rules defining how tasks are accomplished. These rules, referred to as the Environment's World Model, specify the sequence of actions required to achieve specific outcomes. For example, the Environment's World Model might dictate that certain actions (e.g., operating on a receptacle) can only be performed after prerequisite actions (e.g., moving to the receptacle).

Meanwhile, the Agent operates based on its own World Model, which it constructs by interpreting the task and environment prompts. The Agent first determines its high-level reasoning intent—its understanding of what needs to be done—and then selects actions

---

[2]https://huggingface.co/Qwen/Qwen2.5-7B-Instruct

[3]https://huggingface.co/Qwen/Qwen2.5-14B-Instruct

[4]https://huggingface.co/meta-llama/Llama-3.1-8B-Instruct

[5]https://huggingface.co/meta-llama/Llama-3.3-70B-Instruct

according to its internal World Model. However, because the Environment's World Model is manually crafted and may not be fully conveyed through prompts, the Agent's World Model might differ, leading to unexpected behavior. For instance, the Agent might choose an action that aligns with its intent but violates the Environment's rules, or it might misinterpret feedback due to insufficient information from the Environment.

We define a misalignment between the Environment's World Model and the Agent's World Model as a situation where:
- The Environment provides feedback that does not sufficiently clarify its World Model, leaving the Agent unable to adjust its understanding of the rules.

Your task is to analyze the logs from a recent task to determine whether such a misalignment occurred, preventing a fair assessment of the Agent's capabilities. And this misalignment has not been fixed by current 'WrapStep' function. Your analysis will guide us in addressing this issue moving forward.

_______________________________________________

### Experimental Environment Evaluation Template

'''python
{{ experimental_template }}
'''

In this template, the function 'InferRules' is used to define the environment rules. The function 'WrapStep' handles post-processing of the Agent's actions (e.g., splitting them into multiple steps, performing pre-checks, returning more detailed feedback, etc.). This function should not interfere with the Agent's own reasoning. There current implementation is as follows:

'''python
{{ Interface }}
'''

_______________________________________________

### Environment Logs

'''txt
{{ logs }}
'''

Here, each 'Observation' is the feedback returned to the Agent after it executes an action.

_______________________________________________

### Gold Action and Observation Sequence

'''txt
{{ gold_action_obs_sequence }}
'''

_______________________________________________

### Environment Logics and Misalignment Analyzed in the Previous Steps

{{ environment_logics }}

_______________________________________________

### Your Task

Determine whether, during this task, there was a misalignment between the Environment's World Model and the Agent's World Model that hindered a fair assessment of the Agent's capabilities. Choose exactly one of the following outputs:

If there is NO misalignment (i.e., the Agent's failures stem from its own errors or limitations, not a mismatch with the Environment's World Model), output:
<analysis_result> No Misalignment </analysis_result>

If there IS a misalignment (i.e., the Environment's World Model conflicts with the Agent's World Model), output:
<analysis_result> Found Misalignment </analysis_result>
<environment_logic_and_misalignments> the new environment rules and misalignments identified by you, which have not been fixed by current 'WrapStep' function.
</environment_logic_and_misalignments>

The format of the environment logic and misalignment is as follows:
```txt
### Analysis Result 1
Analysis Task ID: xxx
Agent Action Type: xxx # The type of action the Agent attempted to perform, such as "examine", "move object to receptacle", etc.
Agent Action Case: xxx # The specific action the Agent attempted to perform.
Agent High-Level Reasoning Intent: xxx # The Agent's high-level reasoning intent, which may be a general description of the action it was trying to perform.
Environment World Model Rule: xxx # The rule from the Environment's World Model that don't align the Agent's World Model.
Sufficient Environment Feedback: xxx # to offer the Agent adequate information to bridge gaps in understanding the environment's world model. such as "The environment should provide 'xxx' feedback when the Agent attempts to operate on a receptacle without first going to it."
Type: "Bug of current WrapStep function" or "Need to add new logic in the WrapStep function"

### Analysis Result 2
...
```

Note: You should not generate duplicate misalignment analysis results as the ones already provided in the 'Environment Logics and Misalignment Analyzed in the Previous Steps' section.

---

### Analyzer Prompt Template of Experimental Verification

**User message:**
Now you should conduct simulation experiments in the simulator to verify that the environment rules you hypothesized and Misalignment you identified truly exists. You must perform sufficient experiments to confirm or refute your suspicion.

Here are the operations you can use:

1. init_simulator(task_id: str)
- Initializes a new simulator for the specified 'task_id'.
- 'task_id' must be in the format 'int-int' where the first int $\in [0, 5]$.
- The different task types are mapped as follows:

```
0: 'pick_and_place',
1: 'pick_clean_and_place',
2: 'pick_heat_and_place',
3: 'pick_cool_and_place',
4: 'look_at_or_examine_in_light',
5: 'pick_two_obj_and_place'
```

- All subsequent operations occur within this initialized simulator.

2. reset_simulator()
- Resets the current simulator to its initial state.

3. execute_agent_action(agent_action: str)
- Executes an agent action using the 'WrapStep' function.

4. change_last_action_observation(obs: str)
- Updates the last observation returned by the simulator to the specified 'obs'.
- This is useful for simulating the agent's next action in a different environment feedback context.

5. get_next_agent_action()
- Retrieves the next action that the real Agent would perform under the current simulation conditions.
- Note: The Agent's choice of the next action is based on the current environment state, including the outcomes of any previous 'step()' or 'get_next_agent_action()' call, along with the latest observations.

If you believe you have reached a conclusion from your experiments, provide it in this format:

<thought> Your reasoning here </thought>
<environment_logic_and_misalignments> the new environment rules and misalignments identified by you, which have not been fixed by current 'WrapStep' function. </environment_logic_and_misalignments>

The format of the environment logic and misalignment is as follows:
```txt
### Analysis Result 1
Analysis Task ID: xxx
Agent Action Type: xxx # The type of action the Agent attempted to perform, such as "examine", "move object to receptacle", etc.
Agent Action Case: xxx # The specific action the Agent attempted to perform.
Agent High-Level Reasoning Intent: xxx # The Agent's high-level reasoning intent, which may be a general description of the action it was trying to perform.
Environment World Model Rule: xxx # The rule from the Environment's World Model that don't align the Agent's World Model.
Sufficient Environment Feedback: xxx # to offer the Agent adequate information to bridge gaps in understanding the environment's world model. such as "The environment should provide 'xxx' feedback when the Agent attempts to operate on a receptacle without first going to it."
Type: "Bug of current WrapStep function" or "Need to add new logic in the WrapStep function"

### Analysis Result 2
...
```

If you need to carry out more operations in the simulator, respond in the following format, specifying exactly one operation per turn:

<thought> Your reasoning here, you should consider all hypotheses if the simulation result is not as expected </thought>
<action> The single operation you wish to perform (e.g., init_simulator(task_id="x-y"), step(action="x"), execute_agent_action(agent_action="x"), etc.) </action>

Note:
You should verify the correctness of the following, step by step, through your experiments:
1. environment_rules: Use 'execute_agent_action' to confirm that the environment rules you hypothesized are indeed correct, and current 'WrapStep' function is not sufficient.
2. agent_intent_description: Obtain the Agent's intended behavior (e.g., via 'get_next_agent_action') and simulate it by using 'WrapStep' to confirm whether it aligns with your description.
3. identified_misalignment: Through chaning the environment feedback, you can verify whether the misalignment you identified is indeed correct and the environment feedback you hypothesized is indeed sufficient. You can use 'WrapStep' to simulate the agent's action, then use 'change_last_action_observation' to change the environment feedback, and finally use 'get_next_agent_action' to check whether the agent can correctly identify the next action.

---

### Analyzer Prompt Template of Reranking Misalignments Analysis (Ablation Study)

**User message:**
In modern benchmarks evaluating LLM Agent reasoning capabilities, human designers create an Environment with a set of rules defining how tasks are accomplished. These rules, referred to as the Environment's World Model, specify the sequence of actions required to achieve specific outcomes. For example, the Environment's World Model might dictate that certain actions (e.g., operating on a receptacle) can only be performed after prerequisite actions (e.g., moving to the receptacle).

Meanwhile, the Agent operates based on its own World Model, which it constructs by interpreting the task and environment prompts. The Agent first determines its high-level reasoning intent—its understanding of what needs to be done—and then selects actions according to its internal World Model. However, because the Environment's World Model is manually crafted and may not be fully conveyed through prompts, the Agent's World Model might differ, leading to unexpected behavior. For instance, the Agent might choose an action that aligns with its intent but violates the Environment's rules, or it might misinterpret feedback due to insufficient information from the Environment.

We define a misalignment between the Environment's World Model and the Agent's World Model as a situation where:
- The Environment provides feedback that does not sufficiently clarify its World Model, leaving the Agent unable to adjust its understanding of the rules.

Now other human experts have analyzed the logs from a recent task and identified some potential misalignments. Your task is to review these misalignments and choose the most appropriate one.

---

### Experimental Environment Evaluation Template

```python
{{ experimental_template }}
```

In this template, the function 'InferRules' is used to define the environment rules. The function 'WrapStep' handles post-processing of the Agent's actions (e.g., splitting them into multiple steps, performing pre-checks, returning more detailed feedback, etc.). This function should not interfere with the Agent's own reasoning. There current implementation is as follows:

```python
{{ Interface }}
```

________________________________________

### Environment Logs

```txt
{{ logs }}
```

Here, each 'Observation' is the feedback returned to the Agent after it executes an action.

________________________________________

### Gold Action and Observation Sequence

```txt
{{ gold_action_obs_sequence }}
```

________________________________________

### Environment Logics and Misalignment Analyzed in the Previous Steps

{{ environment_logics }}  Note: These logics may not be accurate. They are the environment rules that were previously hypothesized and may contain errors.

________________________________________

### Your Task

Choose the most appropriate misalignment analyzed by human experts from the list below:

{{ new_environment_logics }}

You should respond in format as follows:
```
<review> Your review of each expert output one by one </review>
<expert_id> id of the selected expert output, only the number </expert_id>
```

---

**Optimizer Prompt Template of Interface Generation**

**User message:**
In modern benchmarks evaluating LLM Agent reasoning capabilities, human designers create an Environment with a set of rules defining how tasks are accomplished. These rules, referred to as the Environment's World Model, specify the sequence of actions required to achieve specific outcomes. For example, the Environment's World Model might dictate that certain actions (e.g., operating on a receptacle) can only be performed after prerequisite actions (e.g., moving to the receptacle).

Meanwhile, the Agent operates based on its own World Model, which it constructs by interpreting the task and environment prompts. The Agent first determines its high-level reasoning intent—its understanding of what needs to be done—and then selects actions according to its internal World Model. However, because the Environment's World Model is manually crafted and may not be fully conveyed through prompts, the Agent's World Model might differ, leading to unexpected behavior. For instance, the Agent might choose an action that aligns with its intent but violates the Environment's rules, or it might misinterpret feedback due to insufficient information from the Environment.

We define a misalignment between the Environment's World Model and the Agent's World Model as a situation where:
- The Environment provides feedback that does not sufficiently clarify its World Model, leaving the Agent unable to adjust its understanding of the rules.

Your task is to refine the environment's behavior based on the misalignment identified by the AnalysisAgent, ensuring the Agent's true intentions are executed and its reasoning capabilities are fairly assessed.

---

### Experimental Environment Evaluation Template

```python
{{ experimental_template }}
```

In this template, the function 'InferRules' is used to define the environment rules. The function 'WrapStep' handles post-processing of the Agent's actions (e.g., splitting them into multiple steps, performing pre-checks, returning more detailed feedback, etc.). This function should not interfere with the Agent's own reasoning. There current implementation is as follows:

```python
{{ WrapStep }}
```

---

### Environment Logics and Misalignment Analyzed by AnalysisAgent Previously

{{ last_environment_logics }}

---

### New Environment Logics and Misalignment Analyzed by AnalysisAgent

{{ new_environment_logics }}

---

### Your Task

Based on the misalignments identified by the AnalysisAgent, you need to refine and enhance the 'InferRules' function and 'WrapStep' function to align the Environment's World Model with the Agent's actions and provide clearer feedback. Your output should present the new versions of these functions, ensuring the Agent's high-level reasoning intent is preserved.
Please ensure you follow these requirements:

1. **Function Signature**
The function signature must be:

```python
def InferRules(init_obs, task)
- init_obs: str, the initial observation from the environment, containing all receptacles.
- task: str, the task description.

def WrapStep(env, init_obs, task, agent_action: str, logger)
```

2. **Return Values**
The 'InferRules' function's return value must be a string that describes the environment rules.

The 'WrapStep' function's return value must be three items:
```python
obs: str, reward: bool, done: bool
```

3. **'env.step' Usage**
The only permitted usage pattern for 'env.step' is:
```python
obs, reward, done, info = env.step([agent_action])
obs, reward, done = obs[0], info['won'][0], done[0]
```
No alternative usage forms are allowed. Each call to env.step causes an irreversible change to the environment state; actions must therefore be chosen carefully.

4. **Package Imports**
You may import other packages if necessary, but you must include all imports in your code.

5. **Multiple Calls and Conditional Returns**
You are free to call 'env.step' multiple times or return different 'obs' depending on 'agent_action' or the outcomes of these calls.

6. **You can use logger.debug**
You can use 'logger.debug' to log any information you find useful. The logging will be captured and returned to you in the future for further analysis.

7. Do not modify any aspects not explicitly identified by the AnalysisAgent in the "New Environment Logics and Misalignment Analyzed by AnalysisAgent" section.

8. You must use the following approach when addressing the identified misalignment:
- For each action defined in environment, provide clear, informative, and sufficient feedback from the environment whenever an invalid action is attempted, guiding the Agent toward understanding and adhering to the environment's rules.

9. **Output Format**
You must provide the output strictly in the following format:
<thought>YOUR_THOUGHT_PROCESS_HERE</thought>
YOUR_CODE_HERE

Please ensure your final answer follows these guidelines so that we can accurately bridge the misalignment and allow the environment to execute the Agent's true intentions.

**Optimizer Prompt Template of Experimental Verification**

**User message:**
Now you should conduct simulation experiments in the simulator to verify if the 'InferRules' and 'WrapStep' function you provided is correct for the new environment logics and misalignment analyzed by the AnalysisAgent.

You must perform sufficient experiments to confirm or refute your suspicion. Here are the operations you can use:

1. init_simulator(task_id: str)
- Initializes a new simulator for the specified 'task_id'.
- 'task_id' must be in the format 'int-int' where the first int $\in [0, 5]$.
- The different task types are mapped as follows:

0: 'pick_and_place',
1: 'pick_clean_and_place',
2: 'pick_heat_and_place',
3: 'pick_cool_and_place',
4: 'look_at_or_examine_in_light',
5: 'pick_two_obj_and_place'

- All subsequent operations occur within this initialized simulator.

2. reset_simulator()
- Resets the current simulator to its initial state.

3. execute_agent_action(agent_action: str)
- Executes an agent action using the 'WrapStep' function you generated.

4. change_last_action_observation(obs: str)
- Updates the last observation returned by the simulator to the specified 'obs'.
- This is useful for simulating the agent's next action in a different environment feedback context.

5. get_next_agent_action()
- Retrieves the next action that the real Agent would perform under the current simulation conditions.
- Note: The Agent's choice of the next action is based on the current environment state, including the outcomes of any previous 'step()' or 'get_next_agent_action()' call, along with the latest observations.

6. run_task(task_id: str)
- Runs the entire task in the simulator and returns the running log.
- After running the whole task, you need to call 'init_simulator' or 'reset_simulator' to reinitialize the simulator for further operations.

If you believe you have reached a conclusion from your experiments, provide it in this format:

<thought> Your reasoning here </thought>
<if_need_refine> True/False </if_need_refine>
<refine_strategy> Your strategy for refining the WrapStep function, if if_need_refine is True </refine_strategy>

If you need to carry out more operations in the simulator, respond in the following format, specifying exactly one operation per turn:

<thought> Your reasoning here, you should consider all hypotheses if the simulation result is not as expected </thought>
<action> The single operation you wish to perform (e.g., init_simulator(task_id="x-y"), step(action="x"), execute_agent_action(agent_action="x"), etc.) </action>

---

**Optimizer Prompt Template of Reranking Interface Generation (Ablation Stuty)**

**User message:**
In modern benchmarks evaluating LLM Agent reasoning capabilities, human designers create an Environment with a set of rules defining how tasks are accomplished. These rules, referred to as the Environment's World Model, specify the sequence of actions required to achieve specific outcomes. For example, the Environment's World Model might dictate that certain actions (e.g., operating on a receptacle) can only be performed after prerequisite actions (e.g., moving to the receptacle).

Meanwhile, the Agent operates based on its own World Model, which it constructs by interpreting the task and environment prompts. The Agent first determines its high-level reasoning intent—its understanding of what needs to be done—and then selects actions according to its internal World Model. However, because the Environment's World Model is manually crafted and may not be fully conveyed through prompts, the Agent's World Model might differ, leading to unexpected behavior. For instance, the Agent might choose an action that aligns with its intent but violates the Environment's rules, or it might misinterpret feedback due to insufficient information from the Environment.

We define a misalignment between the Environment's World Model and the Agent's World Model as a situation where:
- The Environment provides feedback that does not sufficiently clarify its World Model, leaving the Agent unable to adjust its understanding of the rules.

Now other human experts have generated a set of code patches to address the misalignment between the Environment's World Model and the Agent's World Model. Your task is to evaluate these patches and select the best one.

---

### Experimental Environment Evaluation Template

```python
{{ experimental_template }}
```

In this template, the function 'InferRules' is used to define the environment rules. The function 'WrapStep' handles post-processing of the Agent's actions (e.g., splitting them into multiple steps, performing pre-checks, returning more detailed feedback, etc.). This function should not interfere with the Agent's own reasoning. There current implementation is as follows:

```python
{{ WrapStep }}
```

---

### Environment Logics and Misalignment Analyzed by AnalysisAgent Previously

{{ last_environment_logics }}

---

### New Environment Logics and Misalignment Analyzed by AnalysisAgent

{{ new_environment_logics }}

_______________________________________________

### Your Task

Choose the best code from the following options to address the misalignment between the Environment's World Model and the Agent's World Model:

{{ code_patches }}

You should respond in format as follows:
```
<review> Your review of each code one by one </review>
<code_id> id of the selected result, only the number </code_id>
```

We present the prompt template of the Vanilla agent in ALFWorld to illustrate the usage of the INFERRULES. For the prompt templates of other agent methods and benchmarks, please refer to the supplemental materials.

---

### Vanilla Agent Prompt Template in ALFWorld

**System message:**
You are an AI assistant solving tasks in a household environment. Your goal is to break down complex tasks into simple steps and plan your actions accordingly.

# Action Space

In this environment, you have a set of high-level actions at your disposal, each corresponding to a typical household activity. These actions are:

- look: look around your current location
- inventory: check your current inventory
- go to (receptacle): move to a receptacle
- open (receptacle): open a receptacle
- close (receptacle): close a receptacle
- take (object) from (receptacle): take an object from a receptacle
- move (object) to (receptacle): place an object in or on a receptacle
- examine (something): examine a receptacle or an object
- use (object): use an object
- heat (object) with (receptacle): heat an object using a receptacle
- clean (object) with (receptacle): clean an object using a receptacle
- cool (object) with (receptacle): cool an object using a receptacle
- slice (object) with (object): slice an object using a sharp object

Although each action may internally consist of multiple embodied steps (e.g., walking to the sink, turning a knob, etc.), from your perspective you need only provide one high-level action at a time.

# Instructions

Single Action per Turn
At each step, you must respond with exactly one action (i.e., the next "thought"). Use the format:

ACTION [object/receptacle specifier]
ACTION [object/receptacle specifier]
For example:
take apple from table
or
go to kitchen.

Environment Feedback
After you provide your single action, the environment will automatically execute it and return the resulting observation. You then decide on your next action based on the updated state.

Reasoning (Chain of Thought)
You may use hidden reasoning to figure out the best next step. However, only output the single action that represents your decision. Do not reveal your entire chain of thought.

Continue Until Task Completion
You will iterate this process—receiving the environment's feedback, deciding on the next action, and outputting a single action—until the task is finished.

# Environment Rule

{InferRules(init_obs, task)}

**User message:**
# Task

{initial_obs}

Begin by examining the environment or taking any initial steps you find relevant. Remember, provide only one action each time.

### E.5  INITIALIZED INTERFACE

Initialized interface we used in ALFWorld:

```
def InferRules(init_obs, task):
    """
    Contains the rules for environment and task execute logic for
    different task types.
    """
    return "There is no rule for this environment."

def WrapStep(env, init_obs, task, agent_action: str, logger):
    """
    Process the agent action and return the next observation, reward,
    and done status.
    """
    obs, reward, done, info = env.step([agent_action])
    obs, reward, done = obs[0], info['won'][0], done[0]
    return obs, reward, done
```

Initialized interface we used in ScienceWorld:

```
def InferRules(init_obs, task):
    """
    Contains the rules for environment and task execute logic for
    different task types.
    """
    return "There is no rule for this environment."
```

```python
def WrapStep(env, init_obs, task, agent_action: str, logger):
    """
    Process the agent action and return the next observation, done
    status and score(returned by the environment).
    """
    obs, _, done, info = env.step(agent_action)
    return obs, done, info["score"]
```

Initialized interface we used in WebShop:

```python
def InferRules(init_obs, task):
    """
    Contains the rules for environment and task execute logic.
    """
    return "There is no rule for this environment."

def WrapStep(env, init_obs, task, agent_action: str, logger):
    """
    Process the agent action and return the next observation, reward,
    and done status.
    """
    obs, reward, done = env.step(agent_action)
    return obs, reward, done
```

Initialized interface we used in M³ToolEval:

```python
def InferRules(task_name, task_type_idx):
    """
    Contains the rules for environment and task execute logic for
    different task types.
    """
    return "There is no rule for this environment."

def WrapStep(env, task_name, instruction, agent_action: str, logger):
    """
    Process the agent action and return the next observation, reward,
    and done status.
    """
    obs, reward, done = env.step(agent_action)
    return obs, reward, done
```

# F   CASE STUDY

## F.1   MISALIGNMENTS ANALYZED BY ANALYZER

We present the misalignments analyzed by Analyzer with Vanilla agent. For the misalignments analyzed by Analyzer with other agent methods, please refer to the supplemental materials.

# # ALFWORLD

| | |
|---|---|
| **Agent Action Type:** | heat object with receptacle |
| **Agent Action Case:** | heat mug 1 with stoveburner 1 |
| **Agent High-Level Reasoning Intent:** | The Agent intended to heat the mug using the stoveburner to fulfill the "put a hot mug in cabinet" task requirement. |
| **Environment Rule:** | The Environment requires heating the mug specifically by the microwave, and the Agent must be at and open the microwave before heating. Heating with the stoveburner or heating without opening the microwave results in no effect. |
| **Sufficient Environment Feedback:** | The environment feedback "Nothing happens." after heating with stoveburner or heating without opening the microwave is insufficient to clarify the correct heating method and prerequisites. |

# # SCIENCEWORLD

| | |
|---|---|
| **Agent Action Type:** | pick up OBJ from CONTAINER / take OBJ from CONTAINER |
| **Agent Action Case:** | pick up orange seed from seed jar, take orange seed from seed jar, take seed from seed jar, pick up seed from seed jar |
| **Agent High-Level Reasoning Intent:** | Agent intends to retrieve a seed from the "seed jar" container using common interaction verbs and syntax ("pick up X from Y", "take X from Y"). |
| **Environment Rule:** | The environment does not support the "take OBJ from CONTAINER" syntax. Furthermore, for the "seed jar", the "pick up OBJ from CONTAINER" syntax is also invalid. The required procedure to access the seeds involves picking up the entire container first ("pick up seed jar") and then likely using a "move" command later. Direct retrieval from the container using "pick up" or "take with from" is disallowed. |
| **Sufficient Environment Feedback:** | The current generic feedback provided by "process_agent_action" for "No known action" is insufficient. Sufficient feedback should diagnose the invalid syntax or procedure, e.g., "The action 'take X from Y' is not valid. To get items from the 'seed jar', try picking up the 'seed jar' first using 'pick up seed jar'." Simulation confirmed this guides the agent correctly. |

# # WEBSHOP

| | |
|---|---|
| **Agent Action Type:** | click |
| **Agent Action Case:** | click[1 ounce (pack of 21)] (or similar option clicks like flavor, color, etc.) |
| **Agent High-Level Reasoning Intent:** | The Agent intended to select a specific product configuration (e.g., size) required by the task before proceeding to purchase or further inspection. |
| **Environment Rule:** | When an Agent clicks on a product option (e.g., size, color, flavor), the internal state of the environment updates to reflect this selection. This selection affects the final product configuration (and potentially price, availability, description shown) when subsequent actions like "Buy Now" or viewing details are taken. The visual representation of the page should ideally reflect this selected state. |
| **Sufficient Environment Feedback:** | The environment currently returns only a confirmation message (e.g., "You have clicked [Option Name]."). This is insufficient as it doesn't show the agent the result of its action in the context of the full page. Sufficient feedback would involve returning the complete observation of the item page *after* the option click, reflecting the updated state (e.g., showing the selected size/flavor visually marked, potentially an updated price, updated product title/description if applicable, and all other page elements like "[Buy Now]"). This allows the agent to verify the selection and its consequences (like price changes) before taking the next step. |

# # M³TOOLEVAL

| | |
|---|---|
| **Agent Action Type:** | web_browsing |
| **Agent Action Case:** | click_url("/about") |
| **Agent High-Level Reasoning Intent:** | The Agent is attempting to use the click_url tool to navigate to the "/about" page by following the clickable link, as per the tool description and the rendered webpage content. |
| **Environment Rule:** | The Environment's World Model expects tool invocations to match the tool name exactly, with arguments provided after the tool name and separated by a comma (i.e., "Action: click_url, /about End Action" instead of "Action: click_url('/about') End Action"). However, the Agent, following the tool signature format shown in the tool descriptions (e.g., click_url(url: str)), invokes the tool as click_url("/about"), which is a common programming convention. |
| **Sufficient Environment Feedback:** | The feedback "Cound not find tool with name click_url('/about')" is insufficient for the Agent to understand the correct invocation format. The environment should provide feedback such as: "Tool arguments should be provided after the tool name, separated by a comma, e.g., 'Action: click_url, /about End Action' instead of 'Action: click_url('/about') End Action'." |

## F.2 INTERFACES GENERATED BY ALIGN

We present the ALIGN-generated interface with Vanilla agent in ALFWorld to illustrate the interface case. For the ALIGN-generated interface with other agent methods and other environments, please refer to the supplemental materials.

ALIGN-generated interface with Vanilla agent in ALFWorld:

```
def InferRules(init_obs, task):
```

```
"""
Provides the rules for environment and task execute logic for
different task types.

Args:
    init_obs: Initial observation string containing information
about the environment
    task: The specific task description

Returns:
    A string describing the environment rules
"""
return """
1. Navigation and Location Rules:
   - You must go to a receptacle before you can examine it, open it,
 close it, or interact with objects in/on it.
   - You can only interact with objects and receptacles that are at
your current location.
   - If you try to interact with a receptacle or object that is not
at your current location, you will be informed that you need to go
to that location first.
   - After successfully going to a location, you are at that
location until you go somewhere else.

2. Object Interaction Rules:
   - To take an object, it must be present at your current location
and visible (not inside a closed receptacle).
   - Once you take an object, it goes into your inventory and is no
longer at its original location.
   - To move an object to a receptacle, you must have the object in
your inventory and be at the target receptacle.
   - To use, heat, clean, cool, or slice objects, you must have the
required objects in your inventory or be at their location.
   - You cannot take an object that is already in your inventory.

3. Container Rules:
   - Some receptacles can be opened and closed (like refrigerators,
microwaves, cabinets, etc.).
   - You must open a closed container before you can take objects
from it or put objects into it.
   - Objects inside closed containers are not visible or accessible
until the container is opened.

4. Action Sequence Requirements:
   - Some tasks require a specific sequence of actions - for example
, to heat food, you need to:
      a) Go to the microwave
      b) Open the microwave
      c) Place the food inside
      d) Close the microwave
      e) Use the microwave
   - The environment will guide you if you're missing a prerequisite
 step for an action.

5. Feedback Interpretation:
   - If an action cannot be performed, the environment will explain
why and what prerequisites are needed.
   - The environment will inform you if you try to take an object
that's already in your inventory.
   - The environment will inform you if you try to move an object
that's not in your inventory.
   - Pay attention to the feedback to understand the current state
of the environment and what actions are possible next.
   - When you successfully go to a location, the environment will
describe what's there.
```

```python
2052        """
2053
2054    def WrapStep(env, init_obs, task, agent_action: str, logger):
2055        """
2056        Process the agent action and return the next observation, reward,
2057        and done status.
2058
2059        Args:
2060            env: The environment object
2061            init_obs: Initial observation string containing information
2062        about the environment
2063            task: The specific task description
2064            agent_action: The action string from the agent
2065            logger: Logger object for debugging information
2066
2067        Returns:
2068            obs: Observation string after the action
2069            reward: Boolean indicating if a reward was received
2070            done: Boolean indicating if the task is complete
2071        """
2072        # Track the agent's current location using an attribute on the env
2073        object
2074        if not hasattr(env, '_current_location'):
2075            env._current_location = None
2076
2077        # Track container states (open/closed) using an attribute on the env
2078         object
2079        if not hasattr(env, '_container_states'):
2080            env._container_states = {}
2081
2082        action_item = {
2083            'matched': False,
2084            'action': None,
2085            'object': None,
2086            'receptacle': None,
2087            'object2': None
2088        }
2089
2090        # Parse the agent action
2091        # Simple actions without parameters
2092        if agent_action.lower() == 'look' or agent_action.lower() == '
2093        inventory':
2094            action_item['matched'] = True
2095            action_item['action'] = agent_action.lower()
2096
2097        # Pattern: go to (receptacle)
2098        elif agent_action.lower().startswith('go to '):
2099            receptacle = agent_action[6:].strip()
2100            action_item['matched'] = True
2101            action_item['action'] = 'go to'
2102            action_item['receptacle'] = receptacle
2103
2104        # Pattern: open/close (receptacle)
2105        elif agent_action.lower().startswith('open ') or agent_action.lower
2106        ().startswith('close '):
2107            action = 'open' if agent_action.lower().startswith('open ') else
2108         'close'
2109            receptacle = agent_action[len(action)+1:].strip()
2110            action_item['matched'] = True
2111            action_item['action'] = action
2112            action_item['receptacle'] = receptacle
2113
2114        # Pattern: take (object) from (receptacle)
2115        elif 'take ' in agent_action.lower() and ' from ' in agent_action.
2116        lower():
```

```
2106                parts = agent_action.split(' from ')
2107                if len(parts) == 2:
2108                    obj = parts[0][5:].strip()   # Remove 'take ' prefix
2109                    receptacle = parts[1].strip()
2110                    action_item['matched'] = True
2111                    action_item['action'] = 'take from'
2112                    action_item['object'] = obj
2113                    action_item['receptacle'] = receptacle

2114            # Pattern: move (object) to (receptacle)
2115            elif 'move ' in agent_action.lower() and ' to ' in agent_action.
        lower():
2116                parts = agent_action.split(' to ')
2117                if len(parts) == 2:
2118                    obj = parts[0][5:].strip()   # Remove 'move ' prefix
2119                    receptacle = parts[1].strip()
2120                    action_item['matched'] = True
2121                    action_item['action'] = 'move to'
2122                    action_item['object'] = obj
2123                    action_item['receptacle'] = receptacle

2124            # Pattern: examine (something)
2125            elif agent_action.lower().startswith('examine '):
2126                something = agent_action[8:].strip()
2127                action_item['matched'] = True
2128                action_item['action'] = 'examine'

2129                # Determine if it's a receptacle or object by checking if it
2130        appears in the initial observation
2131                if something.lower() in init_obs.lower():
2132                    action_item['receptacle'] = something
2133                else:
2134                    action_item['object'] = something

2135            # Pattern: use (object)
2136            elif agent_action.lower().startswith('use '):
2137                obj = agent_action[4:].strip()
2138                action_item['matched'] = True
2139                action_item['action'] = 'use'
2140                action_item['object'] = obj

2141            # Pattern: heat/clean/cool (object) with (receptacle)
2142            elif any(agent_action.lower().startswith(action) for action in ['
        heat ', 'clean ', 'cool ']) and ' with ' in agent_action.lower():
2143                for action in ['heat ', 'clean ', 'cool ']:
2144                    if agent_action.lower().startswith(action):
2145                        parts = agent_action.split(' with ')
2146                        if len(parts) == 2:
2147                            obj = parts[0][len(action):].strip()
2148                            receptacle = parts[1].strip()
2149                            action_item['matched'] = True
2150                            action_item['action'] = action.strip()
2151                            action_item['object'] = obj
2152                            action_item['receptacle'] = receptacle
2153                        break

2154            # Pattern: slice (object) with (object)
2155            elif agent_action.lower().startswith('slice ') and ' with ' in
        agent_action.lower():
2156                parts = agent_action.split(' with ')
2157                if len(parts) == 2:
2158                    obj = parts[0][6:].strip()   # Remove 'slice ' prefix
2159                    obj2 = parts[1].strip()
                    action_item['matched'] = True
                    action_item['action'] = 'slice'
```

```
2160                action_item['object'] = obj
2161                action_item['object2'] = obj2  # Using object2 for the tool
2162        used for slicing
2163
2164        # If action wasn't matched, provide feedback
2165        if not action_item['matched']:
2166            return f"I don't understand the action '{agent_action}'. Please
            use one of the allowed actions from the action space.", False, False
2167
2168        logger.debug(f"Parsed action: {action_item}")
2169
2170        # Get the current observation to check location
2171        test_obs, _, _, _ = env.step(['look'])
2172        test_obs = test_obs[0]
            logger.debug(f"Current observation after 'look': {test_obs}")
2173
2174        # Get inventory to check what objects the agent has
2175        inventory_obs, _, _, _ = env.step(['inventory'])
2176        inventory_obs = inventory_obs[0]
            logger.debug(f"Current inventory observation: {inventory_obs}")
2177
2178        # Improved function to check if an object is in inventory
2179        def is_in_inventory(object_name):
2180            object_name_lower = object_name.lower()
2181            logger.debug(f"Checking if '{object_name_lower}' is in inventory
            ")
2182
2183            # Extract inventory items from the observation
2184            inventory_items = []
2185
2186            # Check for common inventory patterns
2187            if "carrying:" in inventory_obs.lower():
2188                carrying_section = inventory_obs.lower().split("carrying:")
            [1].strip()
2189                inventory_items = [item.strip() for item in carrying_section
            .split(',')]
2190            elif "inventory:" in inventory_obs.lower():
2191                inventory_section = inventory_obs.lower().split("inventory:"
2192            )[1].strip()
2193                inventory_items = [item.strip() for item in
            inventory_section.split(',')]
2194            elif "you are carrying:" in inventory_obs.lower():
2195                carrying_section = inventory_obs.lower().split("you are
2196            carrying:")[1].strip()
2197                inventory_items = [item.strip() for item in carrying_section
2198            .split(',')]
2199
2200            # Also check line by line for inventory items
2201            inventory_lines = inventory_obs.lower().split('\n')
2202            for line in inventory_lines:
2203                line = line.strip()
                if line and not line.startswith(("you are", "carrying:", "
2204            inventory:")):
2205                    inventory_items.append(line)
2206
            logger.debug(f"Extracted inventory items: {inventory_items}")
2207
2208            # Check if object_name or its base name (without numbers) is in
2209        inventory
2210            base_name = ''.join([c for c in object_name_lower if not c.
            isdigit()]).strip()
2211
2212            for item in inventory_items:
2213                # Check for exact match
```

```
2214              if object_name_lower == item or f"{object_name_lower} (in
2215      your inventory)" == item:
2216                  logger.debug(f"Found exact match '{item}' in inventory")
2217                  return True
2218
2219              # Check for base name match (without numbers)
2220              if base_name != object_name_lower and (base_name == item or
2221      f"{base_name} (in your inventory)" == item):
2222                  logger.debug(f"Found base name match '{item}' in
2223      inventory")
                    return True
2224
2225              # Check if item contains the object name
2226              if object_name_lower in item:
2227                  logger.debug(f"Found partial match '{item}' containing
2228      '{object_name_lower}' in inventory")
                    return True
2229
2230              # Check if item contains the base name
2231              if base_name != object_name_lower and base_name in item:
2232                  logger.debug(f"Found partial match '{item}' containing
2233      base name '{base_name}' in inventory")
                    return True
2234
2235          # Direct check for common patterns in the full inventory text
2236          patterns = [
2237              f"carrying: {object_name_lower}",
2238              f"{object_name_lower} (in your inventory)",
2239              f"you are carrying: {object_name_lower}",
2240              f"inventory: {object_name_lower}"
2241          ]
2242
2243          if base_name != object_name_lower:
2244              patterns.extend([
2245                  f"carrying: {base_name}",
2246                  f"{base_name} (in your inventory)",
                    f"you are carrying: {base_name}",
2247                  f"inventory: {base_name}"
2248              ])
2249
2250          for pattern in patterns:
2251              if pattern in inventory_obs.lower():
                    logger.debug(f"Found pattern '{pattern}' in inventory
2252      text")
                    return True
2253
2254          logger.debug(f"'{object_name_lower}' not found in inventory")
2255          return False
2256
2257      # Helper function to check if we're at a location
2258      def is_at_location(location_name):
2259          location_name_lower = location_name.lower()
2260
2261          # If we've already tracked this location, use the tracked value
2262          if env._current_location and location_name_lower in env.
2263      _current_location.lower():
              logger.debug(f"Using tracked location: '{env.
2264      _current_location}'")
2265              return True
2266
2267          # Check if location is mentioned in current observation after "
          You are in"
          if "you are in" in test_obs.lower() and location_name_lower in
      test_obs.lower():
```

```
2268             logger.debug(f"Location '{location_name_lower}' mentioned in
2269      observation after 'You are in'")
2270             return True
2271
2272         # Check if the location is in the first line of the observation
2273      (common format)
2274         first_line = test_obs.split('\n')[0].lower()
2275         if location_name_lower in first_line:
2276             logger.debug(f"Location '{location_name_lower}' found in
2277      first line of observation")
2278             return True
2279
2280         # Check if the observation mentions items at/on the location
2281         location_patterns = [
2282             f"on the {location_name_lower}",
2283             f"in the {location_name_lower}",
2284             f"at the {location_name_lower}"
2285         ]
2286
2287         for pattern in location_patterns:
2288             if pattern in test_obs.lower():
2289                 logger.debug(f"Found pattern '{pattern}' in observation"
2290      )
2291                 return True
2292
2293         logger.debug(f"Not at location '{location_name_lower}'")
2294         return False
2295
2296     # Handle go to action
2297     if action_item['action'] == 'go to':
2298         receptacle = action_item['receptacle']
2299         receptacle_lower = receptacle.lower()
2300
2301         # Check if we're already at this location
2302         if is_at_location(receptacle_lower):
2303             env._current_location = receptacle
2304             return f"You are already at the {receptacle}. You can
2305      interact with it directly.", False, False
2306
2307         # Execute the go to action
2308         obs, reward, done, info = env.step([agent_action])
2309         obs, reward, done = obs[0], info['won'][0], done[0]
2310
2311         # Update the current location if the action was successful
2312         if obs and "nothing happens" not in obs.lower():
2313             env._current_location = receptacle
2314
2315             # If the observation doesn't clearly indicate arrival,
2316      enhance it
2317             if not any(phrase in obs.lower() for phrase in [f"you arrive
2318       at", f"you are at", f"you see {receptacle_lower}"]):
2319                 obs = f"You arrive at the {receptacle}. {obs}"
2320         else:
2321             # Provide more informative feedback
             obs = f"Cannot go to {receptacle}. It might not be a valid
      location or not accessible from here."

         return obs, reward, done

     # Handle examine, open, close, take from, move to actions that
      require being at location
     if action_item['action'] in ['examine', 'open', 'close', 'take from'
      , 'move to']:
         receptacle = action_item['receptacle'].lower() if action_item['
      receptacle'] else ""
```

```
2322            logger.debug(f"Action: {action_item['action']} with receptacle:
2323    {receptacle}")
2324
2325            # Skip location check for examining objects in inventory
2326            if action_item['action'] == 'examine' and action_item['object']
2327    and is_in_inventory(action_item['object']):
2328                # Execute the examine action directly
2329                obs, reward, done, info = env.step([agent_action])
2329                obs, reward, done = obs[0], info['won'][0], done[0]
2330                return obs, reward, done
2331
2332            # Check if we need to be at a receptacle and if we're there
2333            if receptacle and not is_at_location(receptacle):
2333                action_name = action_item['action']
2334                if action_name == 'examine':
2335                    return f"You must go to the {action_item['receptacle']}
2336    first before examining it.", False, False
2337                elif action_name == 'take from':
2338                    return f"You need to go to the {action_item['receptacle
2338    ']} first before taking objects from it.", False, False
2339                elif action_name == 'move to':
2340                    return f"You need to go to the {action_item['receptacle
2341    ']} first before placing objects on/in it.", False, False
2342                else:  # open or close
2343                    return f"You need to go to the {action_item['receptacle
2343    ']} first before you can {action_name} it.", False, False
2344
2345        # Handle open and close actions to track container states
2346        if action_item['action'] in ['open', 'close']:
2347            receptacle = action_item['receptacle']
2348
2348            # Execute the action
2349            obs, reward, done, info = env.step([agent_action])
2350            obs, reward, done = obs[0], info['won'][0], done[0]
2351
2352            # Check for "Nothing happens" and provide more informative
2352    feedback
2353            if obs.strip() == "Nothing happens.":
2354                if action_item['action'] == 'open':
2355                    return f"Unable to open {receptacle}. It might already
2356    be open or not be openable.", reward, done
2357                else:  # close
2358                    return f"Unable to close {receptacle}. It might already
2358    be closed or not be closable.", reward, done
2359
2360            # Update container state tracking
2361            if "successfully" in obs.lower() or "already" in obs.lower():
2362                env._container_states[receptacle.lower()] = 'open' if
2362    action_item['action'] == 'open' else 'closed'
2363
2364            return obs, reward, done
2365
2366        # Check if taking an object that's already in inventory
2367        if action_item['action'] == 'take from':
2367            object_name = action_item['object']
2368            if is_in_inventory(object_name):
2369                return f"You already have the {object_name} in your
2370    inventory. No need to take it again.", False, False
2371
2372        # Check if moving an object that's not in inventory
2373        if action_item['action'] == 'move to':
2373            object_name = action_item['object']
2374            if not is_in_inventory(object_name):
2375                return f"You don't have the {object_name} in your inventory.
2375     You need to take it first.", False, False
```

```python
        # Execute the action in the environment
        logger.debug(f"Executing action in environment: {agent_action}")
        obs, reward, done, info = env.step([agent_action])
        obs, reward, done = obs[0], info['won'][0], done[0]
        logger.debug(f"Environment response: {obs}")

        # Handle special case for "Nothing happens" response
        if obs.strip() == "Nothing happens." and action_item['action'] == '
        take from':
            object_name = action_item['object']
            receptacle_name = action_item['receptacle']

            # Check if it might be because the object is already in
        inventory
            if is_in_inventory(object_name):
                return f"You already have the {object_name} in your
        inventory. No need to take it again.", reward, done

            # Check if it might be because the container is closed
            receptacle_state = env._container_states.get(receptacle_name.
        lower())
            if receptacle_state == 'closed':
                return f"You need to open the {receptacle_name} first before
         taking objects from it.", reward, done

            # Otherwise, the object might not be there
            return f"There is no {object_name} at the {receptacle_name} to
        take. It might be elsewhere or already taken.", reward, done

        # Handle special case for "Nothing happens" response for move action
        if obs.strip() == "Nothing happens." and action_item['action'] == '
        move to':
            object_name = action_item['object']
            receptacle_name = action_item['receptacle']

            # Double-check if the object is in inventory
            if is_in_inventory(object_name):
                # If object is in inventory but move fails, check if
        receptacle is closed
                receptacle_state = env._container_states.get(receptacle_name
        .lower())
                if receptacle_state == 'closed':
                    return f"You need to open the {receptacle_name} first
        before placing objects in it.", reward, done
                else:
                    return f"Unable to move {object_name} to {
        receptacle_name}. Make sure the receptacle is open if it's a
        container.", reward, done
            else:
                # If object is not in inventory, provide clear feedback
                return f"You don't have the {object_name} in your inventory.
         You need to take it first before moving it.", reward, done

        # Handle other "Nothing happens" cases with more informative
        feedback
        if obs.strip() == "Nothing happens.":
            if action_item['action'] == 'open':
                return f"Unable to open {action_item['receptacle']}. It
        might already be open or not be openable.", reward, done
            elif action_item['action'] == 'close':
                return f"Unable to close {action_item['receptacle']}. It
        might already be closed or not be closable.", reward, done
            elif action_item['action'] == 'examine':
                if action_item['object']:
```

```
                    return f"Unable to examine {action_item['object']}. Make
        sure you have it in your inventory or it's visible at your location
        .", reward, done
                else:
                    return f"Unable to examine {action_item['receptacle']}.
        Make sure you're at the right location and it's visible.", reward,
        done
            elif action_item['action'] == 'use':
                return f"Unable to use {action_item['object']}. Make sure
        you have it in your inventory or it's at your current location and
        usable.", reward, done
            elif action_item['action'] in ['heat', 'clean', 'cool', 'slice'
        ]:
                return f"Unable to {action_item['action']} {action_item['
        object']}. Make sure you have all required objects and are at the
        right location.", reward, done
            elif action_item['action'] == 'go to':
                # This case should be handled earlier, but as a fallback
                return f"Cannot go to {action_item['receptacle']}. It might
        not be a valid location in this environment.", reward, done
            else:
                # Generic clarification for other actions
                return f"Action '{agent_action}' resulted in no effect.
        Check if you have all prerequisites or if the action is valid in
        this context.", reward, done

        # For successful move actions, verify the object was actually in
        inventory
        if "successfully" in obs.lower() and "place" in obs.lower() and
        action_item['action'] == 'move to':
            object_name = action_item['object']
            # If the environment says the move was successful, we should
        trust that and not override
            return obs, reward, done

        return obs, reward, done
```