# OpenReview forum: "Agent-Environment Alignment via Automated Interface Generation"
_ICLR.cc/2026/Conference — Submitted to ICLR 2026_

### Official Review · Reviewer_Z6HM · 2025-10-16

**Soundness:** 2
**Presentation:** 2
**Contribution:** 2
**Rating:** 4
**Confidence:** 3

**Summary:**

This paper proposes ALIGN, an automated interface generation framework to mitigate the agent-environment misalignment, a key bottleneck to agent performance. ALIGN has two components: *InferRule* (automatically generated static rules) and *WrapStep* (step-wise observation augmentation with feedback). A failure-driven analyzer and optimizer loop learns interfaces and evaluates agents in the wrapped environments. The reported results show improvements across three domains against baselines.

**Strengths:**

1. The paper points out the agent-environment misalignment as a key bottleneck for agent performance.
2. The implementation is straightforward and does not need to change the code of agent or environment, only adding the interface layer.

**Weaknesses:**

1. The interface construction is demonstrated only in three domains as provided in Appendix, but does not discuss a unified design space for interface representation, limiting generalization beyond tested settings.
2. Only donwstream success rate and proportion ofconsecutive invalid actions are reported for measuring effectiveness. However, it is unclear whether interfaces are learned only on training tasks and then frozen for test. Otherwise, the analyzer and optimizer loop could constitute a test-time adaptation. Moreover, *WrapStep* appears to provide tailored corrective hints which may make the comparisons unfair.
3. The limitations of the method are not properly mentioned.

**Questions:**

1. Were interfaces generated only on training tasks and frozen for test? Please specify any test-time adaptation.
2. Can you provide a unified design space for interface representation?
3. Can you add an ablation with limit information provided (interface that avoid tailored corrective hints) to test robustness?
4. What are the limitations of ALIGN and what are common failure cases?

---

> ### Author Response · Authors · 2025-11-23
>
> Thank you for your valuable review and detailed suggestions!
>
> ## Weakness 1 & Question 2
>
> > Unified design space for interface representation.
>
> In the revised manuscript (Section 3.1), we have provided a formal definition of the interface as $\Phi = \langle f_{\text{info}}, f_{\text{act}}, f_{\text{obs}} \rangle$, which is **domain-agnostic and applies beyond text-based discrete action spaces**. This formalization establishes a principled design space for interface representation. And the core insight, that misalignment arises from information loss in how environments present constraints ($f_{info}$) and state transitions ($f_{obs}$) to agents, is fundamental and extends to various interaction modalities.
>
> In the meantime, we have discussed extensions to the full design space in the Limitations and Future Work section, including automatically generating and optimizing $f_{\text{act}}$ to bridge the gap between an agent's preferred action representation and the environment's expected format remains an important direction.
>
> ## Weakness 2 & Question 1
>
> > Experiment setting declaration
>
> **Yes, interfaces are generated only on training tasks and remain frozen during testing.** There is no test-time adaptation. All reported results are measured on held-out test sets with frozen interfaces. This ensures fair comparison between w/o ALIGN and w/ ALIGN conditions.
>
> We have explicitly stated this in Section 4.1 that "During the interface generation and refinement process, only tasks from the training set are used. The interface logic is fixed and remains unchanged during testing." in the revised manuscript.
>
> ## Weakness 2 & Question 3
>
> > Discussion of WrapStep
>
> We apologize for the misunderstanding. Nevertheless, we would like to clarify that incorporating WrapStep does not result in an unfair comparison.
>
> The purpose of WrapStep is to augment the environment's originally insufficient feedback into complete observations that accurately convey state transitions to the agent. As we argue throughout the paper, **agent-environment misalignment fundamentally arises from information loss**, the environment fails to adequately communicate its state transitions and the reasons for state transitions. WrapStep addresses this by transforming raw, ambiguous observations into informative feedback that better conveys the actual state transitions and their causes.
>
> Our goal is not to achieve SOTA through "helping" the agent, but to demonstrate that **much of the performance gap attributed to agent reasoning deficiencies actually stems from poor interface design**. The comparison is fair because we are measuring the effect of improving the communication channel between agent and environment, not providing privileged information unavailable to the environment itself. All feedback provided by WrapStep is derived from the environment's actual state and rules, and WrapStep automatically makes the implicit information explicit.
>
> The comparison between w/o ALIGN and w/ ALIGN measures whether better interfaces enable agents to perform tasks they were previously capable of but failed due to miscommunication, which is precisely the research question we aim to address.
>
> ## Weakness 3 & Question 4
>
> > Limitations and common failure cases
>
> We have added a comprehensive Limitations and Future Work section at the end of the main text, discussing three key directions: (1) extension to more diverse and complex environments, (2) automatic generation of f_act beyond f_info and f_obs, and (3) more comprehensive metrics for interface quality.
>
> Regarding common failure cases after alleviating misalignment: The main reason agents still fail after solving agent-environment misalignment lies in **whether the agent itself possesses the capability to complete the tasks**. In other words, after the agent can fully understand the current environment state (after solving agent-environment misalignment), whether it can make correct decisions. For example, in ScienceWorld, tasks are more complex than ALFWorld, requiring exploration across multiple rooms and using various actions in a larger action space to complete tasks. A common error is that due to the agent's poor long-chain reasoning ability, it may repeat previous actions after multiple unsuccessful exploration attempts to find an item, ultimately leading to failure. For WebShop, most failures are due to the agent's inability to browse all items and select items that fully comply with instructions for purchase, or inability to perform re-search operations. Designing better memory and planning modules might solve these problems, which is also one of the current research directions in agent work.
>
> This is also what we mentioned in the paper: solving agent-environment misalignment helps discover other factors that truly affect agent performance. Applying ALIGN-generated interfaces can help agents unleash their true capabilities, but cannot help them solve tasks they are inherently unable to handle.

---

> ### Author Response · Authors · 2025-11-27
> **Follow-up on Our Rebuttal**
>
> Dear Reviewer,
>
> Thank you for your valuable feedback. We have provided detailed responses to all concerns raised. With the deadline approaching, we would greatly appreciate any comments on our responses to ensure we've addressed your questions satisfactorily. Please feel free to reach out if you have any further questions.

---

### Official Review · Reviewer_VG1Y · 2025-10-20

**Soundness:** 3
**Presentation:** 3
**Contribution:** 3
**Rating:** 6
**Confidence:** 4

**Summary:**

The paper introduces ALIGN, an automated interface-generation framework to address Agent–Environment Misalignment (AEM). ALIGN operates as a lightweight wrapper without modifying either the agent or the environment. Across ALFWorld, ScienceWorld, WebShop, and M3ToolEval, it delivers substantial gains for diverse agent paradigms (Vanilla, ReAct, Self-Consistency, Self-Refine, Planning). Interfaces produced in a single pass transfer across different agent algorithms and LLM backbones, and in multiple comparisons outperform manually engineered interfaces.

**Strengths:**

1. The paper clearly defines the problem and is well motivated, identifying agent–environment misalignment as a key, underexplored bottleneck and substantiating its impact with preliminary evidence, which provides strong motivation for the work.

2. The methodology is novel: unlike manually engineered interfaces, ALIGN automatically detects misalignments and synthesizes interfaces, introducing the notion of an auto-aligned interface (Sec. 3).

3. The empirical study is substantial, demonstrating consistent gains across four benchmarks and five agent families, and the ablations reasonably verify the contribution of each component.

**Weaknesses:**

1. The experiments are conducted in environments with discrete, text-based action spaces. It is less clear how the ALIGN framework would scale to environments with significantly more complex, high-dimensional, or continuous observation/action spaces; It is less clear how the ALIGN framework would scale to environments with significantly more complex, or continuous observation/action spaces.

2. The dependence on high-capability models. The Analyzer and Optimizer currently rely on top proprietary LLMs, raising concerns about accessibility and reproducibility. This raises questions about the accessibility of the framework and its performance when using smaller, open-source models for these roles. An experiment or discussion on this aspect would enhance the paper's contribution.

**Questions:**

I have a few questions and suggestions that I hope will help clarify some aspects of the work and further strengthen the paper.

1. How about the generalization to more complex environments? The experiments are convincingly demonstrated in text-based settings with discrete action spaces. It would be helpful to include a discussion of the framework’s applicability to more complex or continuous observation/action spaces.

2. The success of the Analyzer and Optimizer seems intrinsically linked to the advanced reasoning and coding capabilities of the proprietary models used.Have you explored using smaller or open-source models (e.g., Llama-3.1-70B, Qwen2.5-14B) for the core components of ALIGN? How does the quality of the generated interface degrade when using less capable models?

3. Comparison with agent fine-tuning methods. The paper frames ALIGN as a training-free approach that modifies the interface rather than the agent. However, an alternative is to use failed trajectories as training data to fine-tune the agent itself (e.g., DPO). A direct experimental comparison against a fine-tuning baseline using the same error signals would help demonstrate the superiority and efficiency of the proposed method.

---

> ### Author Response · Authors · 2025-11-23
>
> Thank you for your supportive review and suggestions!
>
> ## Weakness 1 & Question 1
>
> > Generalization to more complex environments.
>
> We acknowledge that our current evaluation focuses on environments with discrete, text-based action spaces, and we appreciate the opportunity to discuss the framework's potential applicability to more complex settings.
>
> In the revised manuscript (Section 3.1), we have provided a formal definition of the interface as $\Phi = \langle f_{\text{info}}, f_{\text{act}}, f_{\text{obs}} \rangle$, which is **domain-agnostic and applies beyond text-based discrete action spaces**. The core insight, that misalignment arises from information loss in how environments present constraints ($f_{info}$) and state transitions ($f_{obs}$) to agents, is fundamental and extends to various interaction modalities.
>
> For environments with continuous or high-dimensional observation/action spaces, the manifestation of misalignment would differ but the underlying principle remains: agents need sufficient information to understand state transitions and action effects. For example:
> - In continuous control tasks, $f_{obs}$ might need to clarify why a gripper failed to grasp an object (e.g., "grip force insufficient" vs. "object too far"), rather than just returning a generic failure signal.
> - In vision-based GUI agents, $f_{info}$ might need to expose interaction constraints (e.g., "buttons are only clickable when enabled"), while $f_{obs}$ could augment visual feedback with structured state descriptions.
>
> We have added the application in more complex environments as an important direction for future work in the Limitations and future work section. We believe the formal interface framework established in this work provides a foundation for such extensions, though empirical validation in these domains remains valuable future work.
>
> ## Weakness 2 & Question 2
>
> > Using smaller or open-sourced models.
>
> To validate the feasibility of using open-source models, we conducted experiments using Qwen2.5-32B-Instruct for the Analyzer and Optimizer components.
>
> **Experiment 1: Qwen2.5-32B-Instruct as both Analyzer and Optimizer**
>
> We used Qwen2.5-32B-Instruct for both components to optimize a Vanilla Agent (base model: Qwen2.5-7B-Instruct) on ALFWorld:
>
> | Iter0 | Iter1 | Iter2 | Iter3 | Iter4 | Iter5 | Iter6 | Iter7 |
> |-------|-------|-------|-------|-------|-------|-------|-------|
> | 13.43 | 5.22 | 8.96 | 11.94 | 5.22 | 6.72 | 6.72 | 0.00 |
>
> The results show significant degradation. Analysis reveals that Qwen2.5-32B-Instruct as Analyzer frequently identifies incorrect misalignments. For example, in the case from Figure 1, it fails to identify the underlying environment logic (the precondition that "go to" must precede "examine"), instead incorrectly attributing the error to the environment not returning examination results.
>
> **Experiment 2: Qwen2.5-32B-Instruct as Optimizer only**
>
> We tested Qwen2.5-32B-Instruct's ability to generate interface code based on misalignments correctly identified Analyzer:
>
> |       | Task-success rate |
> |-------|-------------------|
> | Before optimization | 29.85% |
> | Gemini 2.5 Pro (Optimizer) | 42.54% |
> | Qwen2.5-32B-Instruct (Optimizer) | 29.85% |
>
> Qwen2.5-32B-Instruct failed to generate effective interface modifications, leaving performance unchanged.
>
> **Analysis:** The poor performance of Qwen2.5-32B-Instruct stems from two key limitations: (1) limited reasoning capability under long contexts for analyzing complex agent-environment interactions, and (2) insufficient code understanding and generation capability for modifying substantial code blocks.
>
> These results suggest that current open-source models do not yet possess the necessary reasoning and coding capabilities for the Analyzer and Optimizer roles. However, we note that this is a rapidly evolving landscape—as open-source models continue to improve in reasoning and coding capabilities, they may become viable alternatives. Importantly, even when open-source models cannot generate interfaces themselves, they can still benefit from interfaces generated by more capable models, as demonstrated in our cross-model generalization experiments (Table 3b). This enables practitioners to leverage the advantages of open-source models (private deployment, edge computing, lower costs) while still obtaining performance improvements through ALIGN-generated interfaces.

---

> > ### Author Response · Authors · 2025-11-23
> > **Continued**
> >
> > ## Question 3
> >
> > > Comparison with agent fine-tuning methods.
> >
> > We would like to clarify the purpose and advantages of ALIGN compared to agent fine-tuning approaches.
> >
> > **Purpose of ALIGN:** ALIGN's goal is to systematically reduce agent-environment misalignment by automatically generating interfaces, thereby improving the performance of existing agents and enabling agent developers to focus on agent method design rather than environment-specific engineering. This is fundamentally different from achieving SOTA performance through agent training.
> >
> > **Key advantages over fine-tuning methods:** Compared to training-based approaches like DPO, ALIGN offers two critical advantages:
> >
> > 1. **Generalization across agents and models:** As demonstrated in Section 4.4, ALIGN-generated interfaces generalize to: (a) different agent architectures without regenerating the interface (Table 3a), and (b) larger and heterogeneous LLMs across model families (Table 3b). A single interface generation effort benefits multiple agent implementations.
> > 2. **Generality to both trainable and non-trainable agents:** ALIGN is effective for closed-source LLMs (e.g., GPT-4.1, Table 4a) where fine-tuning is not accessible, and even for domain-specific models already trained on the environment (e.g., GiGPO-Qwen2.5-7B-Instruct, Table 4b). Notably, **even agents trained with methods like DPO still benefit from ALIGN-generated interfaces** (Table 4b shows improvement on an RL-trained model), indicating that misalignment persists even after training and that interface improvement is complementary to agent training.
> >
> > In essence, fine-tuning addresses "how can *this specific agent* perform better in *this environment*," while ALIGN addresses "how can we make *any agent* interact more effectively with *this environment*." These are complementary approaches: ALIGN provides a one-time infrastructure improvement that benefits all current and future agents, whereas fine-tuning must be repeated for each new agent or model update.

---

> ### Author Response · Authors · 2025-11-27
> **Follow-up on Our Rebuttal**
>
> Dear Reviewer,
>
> Thank you for your valuable feedback. We have provided detailed responses to all concerns raised. With the deadline approaching, we would greatly appreciate any comments on our responses to ensure we've addressed your questions satisfactorily. Please feel free to reach out if you have any further questions.

---

### Official Review · Reviewer_A53P · 2025-10-30

**Soundness:** 3
**Presentation:** 2
**Contribution:** 3
**Rating:** 4
**Confidence:** 3

**Summary:**

This paper formalizes a problem: the discrepancy between an agent's internal expectations about its actions and the environment’s actual transition dynamics (the agent-environment misalignment problem). This problem significantly reduces the agent's task success rate.

Some recent studies have addressed this issue by manually designing interfaces. However, this approach is very labor-intensive, and its optimality for the agent cannot be guaranteed.

In this paper, the authors propose the ALIGN framework, which automatically generates intermediate interfaces to alleviate this misalignment without modifying the agent's logic or the environment code. The ALIGN framework autonomously analyzes failure trajectories, infers interface rules, verifies these rules through environment interactions, and iteratively optimizes the alignment effect. Experiments on four benchmark sets (ALFWorld, ScienceWorld, WebShop, and M3ToolEval) demonstrate that ALIGN achieves consistent improvements across various LLM agent architectures (e.g., ReAct, Self-Consistency, and Planning).

**Strengths:**

(1) The work introduces a new problem formulation: agent-environment misalignment, which captures failure modes overlooked by prior works focusing solely on reasoning or reward modeling.

(2) The authors conducted extensive experiments on multiple benchmarks and agent frameworks, providing strong empirical evidence for the robustness and generalization capability of the proposed approach.

**Weaknesses:**

(1) Figure 1 is difficult to understand and is not well aligned with the examples in lines 100–103 (INFERRULES and WRAPSTEP). In addition, some figures could benefit from clearer captions.

(2) While this paper elucidates the functionality and semantic boundaries of the Interface through descriptive language in several places, it does not provide a formal definition. This limitation hinders a more precise analysis of the agent–environment misalignment problem and a deeper understanding of the ALIGN Framework. The authors are encouraged to provide an explicit definition of the Interface and to conduct a more detailed analysis of the agent–environment misalignment problem from the interface perspective, including a discussion of prior works that exhibit such misalignment.

(3) Although the ablation results are reported, the distinction between INFERRULES and WRAPSTEP in terms of their relative contributions could be made clearer. Providing a per-component breakdown or a visualization of the learned rule structures would further improve interpretability.

**Questions:**

See Weakness

If the author is willing to explain/solve these issues, I am willing to raise my score.

---

> ### Author Response · Authors · 2025-11-23
>
> Thank you for your valuable review and detailed suggestions!
>
> ## Weakness 1
>
> > Figure 1 is difficult to understand and is not well aligned with the examples in lines 100–103 (INFERRULES and WRAPSTEP). In addition, some figures could benefit from clearer captions.
>
> We recognize that the "Information Alignment" and "Interaction Alignment" terminology in the previous version of Figure 1 was potentially confusing, and that directly using the notation from our method would be more clear and direct. We have redesigned the right panel of Figure 1 to directly illustrate the functionality of InferRules and WrapStep and the phases in which they operate, using the concrete case from the figure. We have also revised the caption of Figure 1 accordingly.
>
> Following your suggestion, we have also enriched the caption of Figure 2 to provide detailed explanations of the functionality of the InferRules and WrapStep modules. We believe these revisions substantially improve the clarity and accessibility of our method's presentation.
>
> ## Weakness 2
>
> > Formal definition of interface and detailed analysis of the agent–environment misalignment problem from the interface perspective.
>
> We have added a complete formal definition of Interface in Section 3.1 of the revised manuscript. We formally define the interface as a tuple of mapping functions: $\Phi = \langle f_{\text{info}}, f_{\text{act}}, f_{\text{obs}} \rangle$, and analyze the root causes of agent-environment misalignment from the interface perspective, identifying two primary mechanisms: (1) Under-specified Constraints due to inadequate $f_{\text{info}}$, and (2) State Aliasing via Lossy Observations due to inadequate $f_{\text{obs}}$. This formalization provides a principled foundation for understanding why misalignment occurs and how our method addresses it.
>
> > including a discussion of prior works that exhibit such misalignment
>
> While some prior works have proposed environment-specific interface improvements (e.g., SWE-agent [1] for coding tasks, Agent S [2] for GUI agents), they focus on manually designing better interfaces for particular domains rather than systematically studying agent-environment misalignment as a general phenomenon or providing automated solutions.
>
> Our work empirically demonstrates that **agent-environment misalignment is pervasive** across diverse environments, agent architectures, and model types, from open-source models to closed-source models to domain-specific trained models.
>
> [1] Yang, Jimenez, et al. "SWE-agent: Agent-computer interfaces enable automated software engineering."
>
> [2] Agashe, Han, et al. "Agent S: an open agentic framework that uses computers like a human."
>
> ## Weakness 3
>
> > Although the ablation results are reported, the distinction between INFERRULES and WRAPSTEP in terms of their relative contributions could be made clearer. Providing a per-component breakdown or a visualization of the learned rule structures would further improve interpretability.
>
> We have provided clearer explanations of the roles and mechanisms of InferRules and WrapStep in the revised manuscript (lines 216-227 and Figure 2), as well as their relationship to the root causes of misalignment (lines 189-196).
>
> InferRules and WrapStep are complementary mechanisms designed to address agent-environment misalignment at different stages of interaction, and they may target the same underlying misalignment through different means. Using the case in Figure 1 as an example: to resolve the discovered misalignment, InferRules provides additional implicit constraints at agent initialization (e.g., "Navigation Rules: You must go to receptacle before..."), while WrapStep detects when the agent violates these action rules and provides explicit corrective feedback explaining why the action failed and the state did not transition. We designed the ablation study (Table 5) to validate the effectiveness of both modules.
>
> The reason WrapStep shows more pronounced effects is that we observe current LLMs are more strongly influenced by recent prompts (i.e., step-wise observations from WrapStep) in multi-turn interactive tasks. In contrast, even when additional constraints are provided in the system prompt (via InferRules), agents may fail to consistently follow this information ($\tilde{\mathcal{I}}$) due to limitations in their ability to maintain and apply long-context constraints throughout extended interactions.
>
> Thank you again for your detailed suggestions!

---

> ### Author Response · Authors · 2025-11-27
> **Follow-up on Our Rebuttal**
>
> Dear Reviewer,
>
> Thank you for your valuable feedback. We have provided detailed responses to all concerns raised. With the deadline approaching, we would greatly appreciate any comments on our responses to ensure we've addressed your questions satisfactorily. Please feel free to reach out if you have any further questions.

---

### Official Review · Reviewer_adFX · 2025-11-02

**Soundness:** 2
**Presentation:** 2
**Contribution:** 2
**Rating:** 4
**Confidence:** 4

**Summary:**

This paper identifies and tackles the overlooked challenge of agent-environment misalignment in interactive decision-making tasks with LLM-based agents. While most prior work focuses on improving agent reasoning or environment design, this paper pinpoints interface design, the communication layer between the agent and environment, as a bottleneck. The authors propose ALIGN, an automated framework that generates interface wrappers, enriching environment descriptions and step-wise observations to better inform the agent. The paper shows that ALIGN improves performance substantially, generalizes across agents and LLMs, and outperforms manually designed interfaces. It is evaluated on four benchmarks spanning embodied AI, web navigation, and tool-use.

**Strengths:**

1. This paper identifies a real and underexplored problem of agent-environment misalignment and shows through motivating examples and pilot studies (e.g., "Examine bookshelf"->“Nothing happens” in ALFWorld) that it significantly limits agent performance.
2. ALIGN introduces a plug-and-play prompting wrapper approach that does not require changing agent logic or environment code.
3. ALIGN is evaluated on 4 benchmarks, 5 agent methods, and multiple LLMs. The performance improvements seem to be large and consistent (e.g., up to +57.46% on ALFWorld).

**Weaknesses:**

1. Ideal Assumption: The method assumes that environment rules are fully available to agents, which may not always be the case in real-world settings.

2. Potential Unfair Comparison: In the Analyzer Prompt Template for Misalignment Analysis, the Gold Action and Observation Sequence are provided, which is often unavailable. This causes unfair comparison with other baselines that do not have access to them.

3. Potential Unfair Comparison: In the Implementation Detail section, it is mentioned that Qwen2.5-7B-Instruct is used as the base model, while Gemini 2.5 and GPT-4.1 Pro are leveraged as the Optimizer and Analyzer. This also causes unfair comparison with other baselines that only use Qwen2.5-7B-Instruct.

4. Besides, the paper lacks analysis on the detection rate of Agent-Environment Misalignment. Are there different types or causes of such misalignment? If so, could the authors provide a taxonomy?

5. The authors mainly perform evaluation on Qwen2.5-7B-Instruct and GPT-4.1. How effective is the proposed method on more advanced LLMs (such as GPT-5, Gemini-2.5-Pro, etc.) and on reasoning LLMs with varying reasoning capabilities?

6. The experiments show the iteration results without verification but lack iteration results with verification, which are also important to fully understand the method’s effectiveness.

7. In addition, cost and time analysis for different iterations and components are not provided.

8. It is also encouraged to report the variance or standard deviation of the main results.

**Questions:**

See the weakness section above

---

> ### Author Response · Authors · 2025-11-23
>
> Thank you for your valuable review and detailed comments!
>
> ## Weakness 1
>
> > Ideal Assumption
>
> Thank you for this comment. You mentioned that our method assumes "environment rules are fully available to agents." We would like to clarify that our work is actually motivated by precisely the opposite observation. As formalized in Section 3.1, environments present agents with an environment foundational information description ($\mathcal{I}$), which is a standard component in virtually interactive environments. However, **it is precisely because environment rules are NOT fully presented to agents that the agent-environment misalignment problem we address arises**, ultimately leading to task failures.
>
> The fundamental assumption of our work is that **the observations returned to agents are often insufficient to accurately represent the true state transitions**, which gives rise to agent-environment misalignment. Both our preliminary experiments (Appendix C.1) and the experiments presented in Section 4 empirically validate this assumption.
>
> Our method, ALIGN, does not assume that environment rules are pre-known. Rather, it automatically discovers these implicit rules through iterative analysis of failed agent trajectories and generates interfaces that make these rules explicit. This is fundamentally different from assuming rules are available. We are *inferring and exposing* previously hidden rules. Furthermore, while our validation is conducted in simulated environments, this assumption holds equally in real-world settings. For example, in robotics scenarios, if a robot agent fails to operate an induction cooker, the system should determine how to present relevant information (e.g., instruction manual excerpts) to enable the agent's next decision. In our setup, ALFWorld serves as a simulated instantiation of such real-world environments. We acknowledge that validation in real-world environments remains important future work and have added this point to our revised Future work section.
>
> ## Weakness 2 & 3
>
> > Potential Unfair Comparison
>
> Thank you for raising this concern. We would like to clarify that there is no unfair comparison in our work.
>
> In this paper, we emphasize that after using ALIGN-generated interfaces to alleviate agent-environment misalignment, the original agent methods can achieve performance improvements **without modifying agent logic or environment code**. During testing with ALIGN-generated interfaces, agents do **not** have access to the gold actions and use the same model configurations (e.g., Qwen2.5-7B-Instruct) as in the w/o ALIGN setting. Therefore, the comparison between w/ ALIGN and w/o ALIGN is fair and controlled.
>
> The gold action and observation sequences are provided **only during the interface generation phase** (which uses the training set) to help the Analyzer more efficiently identify misalignments. This is analogous to supervised learning where training data contains labels, but the learned model is evaluated on test data without labels. **Importantly, once the interface is generated, it is fixed and applied to all test cases where agents have no access to gold trajectories.**
>
> Furthermore, we acknowledge that we did not clearly state in the original manuscript that for WebShop and M$^3$ToolEval environments, no Gold Action and Observation Sequence is provided even during interface generation. We have clarified this setting in the revised version (see Appendix E.4). This demonstrates the generality of our method: **even without access to ground-truth trajectories, the ALIGN Analyzer can successfully identify misalignments through iterative analysis of failed agent interactions.**

---

> > ### Author Response · Authors · 2025-11-23
> > **Continued**
> >
> > ## Weakness 4
> >
> > > Besides, the paper lacks analysis on the detection rate of Agent-Environment Misalignment. Are there different types or causes of such misalignment? If so, could the authors provide a taxonomy?
> >
> > We acknowledge that quantifying "detection rate" would require a ground-truth set of all misalignments in an environment, which is challenging to establish. In our experiments, we observe that the algorithm typically converges because no new misalignments are identified (condition 3) rather than hitting the maximum iteration limit. For example, in ALFWorld, the algorithm converged after 7 iterations when no new misalignments could be discovered from failed trajectories, as shown in the discussion of Weakness 6, suggesting that the major misalignments affecting agent performance have been addressed.
> >
> > Regarding the taxonomy of agent-environment misalignment, we have added a formal analysis in Section 3.1 of the revised version. Misalignment primarily arises through two mechanisms: Under-specified Constraints (inadequate $f_{info}$) and State Aliasing via Lossy Observations (inadequate $f_{obs}$).
> >
> > These manifest in three common patterns:
> > 1. Missing action rule feedback: agents receive failure observations without clear corrective guidance when violating execution rules;
> > 2. Incomplete state transition representation: the environment fails to clearly convey state transitions;
> > 3. Insufficient or unclear observations: observations lack detail to support informed decision-making.
> >
> > We also provide concrete examples from each benchmark in Appendix F.1, demonstrating how ALIGN addresses these different types of misalignment.
> >
> > ## Weakness 5
> >
> > > The authors mainly perform evaluation on Qwen2.5-7B-Instruct and GPT-4.1. How effective is the proposed method on more advanced LLMs (such as GPT-5, Gemini-2.5-Pro, etc.) and on reasoning LLMs with varying reasoning capabilities?
> >
> > Thank you for this question. We have conducted experiments on GPT-5 using the interface generated with gpt-4.1-mini-2025-04-14 as the base model mentioned in Section 4.4. The results on ALFWorld are:
> >
> > |  | Task-success rate |
> > |---|---|
> > | w/o ALIGN | 83.58% |
> > | w/ ALIGN | 90.30% (+6.72%) |
> >
> > Due to cost constraints, we have not tested on Gemini-2.5-Pro. However, these results, combined with Table 4, demonstrate that agent-environment misalignment exists broadly across open-source models (Qwen series), domain-specific models (GiGPO), closed-source reasoning models (GPT-5), and closed-source non-reasoning models (GPT-4.1 series). ALIGN-generated interfaces consistently alleviate this problem and improve agent performance across all these model types.
> >
> > ## Weakness 6
> >
> > > The experiments show the iteration results without verification but lack iteration results with verification, which are also important to fully understand the method’s effectiveness.
> >
> > Thank you for this suggestion. We have added the iteration results with full verification on ALFWorld. Combined with Table 6, these results demonstrate that Experimental Verification effectively mitigates LLM hallucination issues, enabling stable identification of agent-environment misalignments and successful interface generation.
> >
> > | Iter0 | Iter1 | Iter2 | Iter3 | Iter4 | Iter5 | Iter6 | Iter7 |
> > |-------|-------|-------|-------|-------|-------|-------|-------|
> > | 13.43 | 39.55 | 46.27 | 50.75 | 55.97 | 62.69 | 63.43 | 62.69 |

---

> ### Author Response · Authors · 2025-11-23
> **Continued**
>
> ## Weakness 7
>
> > In addition, cost and time analysis for different iterations and components are not provided.
>
> Due to space constraints, we have provided a detailed token consumption analysis in Appendix D, including the average token consumption per iteration for the Analyzer, Optimizer, and the overall system. We have also discussed strategies for reducing costs in future work. Since the logic is consistent across iterations, the cost per iteration remains relatively stable.
>
> We apologize for not recording execution time during our experiments. However, we can provide a rough estimate based on the method logic. For ALFWorld, each iteration requires: (1) running experiments on 18 selected training tasks (this can be parallelized, with speed depending on agent deployment resources), and (2) 5-20 LLM API calls for analysis and interface generation.
>
> Importantly, **the token cost and time cost of interface generation is a one-time expense**, not a recurring cost for each task execution. Once generated, the interface is reused across all test cases and generalizes to different agents. Therefore, from an amortization perspective, the method's cost becomes increasingly economical as the environment is utilized more frequently, with the one-time interface design cost distributed across multiple uses and becoming proportionally smaller with increased usage.
>
> ## Weakness 8
>
> > It is also encouraged to report the variance or standard deviation of the main results.
>
> Thank you for this suggestion. We conducted 3 independent runs to assess result stability during evaluation. The variance across runs is reported below:
>
> |                  | ALFWorld | ScienceWorld | WebShop | M$^3$ToolEval |
> | :--------------- | :------: | :----------: | :-----: | :-----------: |
> | Vanilla          | 0.000000 |  0.204265    | 0.000000 |  0.000000    |
> | ReAct            | 0.000050 |  0.217538    | 0.000089 |  0.000043    |
> | Self-Consistency | 0.000037 |  0.197378    | 0.000140 |  0.000043    |
> | Self-Refine      | 0.000161 |  0.155519    | 0.000000 |  0.000043    |
> | Planning         | 0.000012 |  0.062370    | 0.000089 |  0.000129    |
>
> The results show low variance across most benchmarks under the temperature=0.0 setting, confirming the stability and reproducibility of our findings.

---

> ### Author Response · Authors · 2025-11-27
> **Follow-up on Our Rebuttal**
>
> Dear Reviewer,
>
> Thank you for your valuable feedback. We have provided detailed responses to all concerns raised. With the deadline approaching, we would greatly appreciate any comments on our responses to ensure we've addressed your questions satisfactorily. Please feel free to reach out if you have any further questions.

---

### Author Response · Authors · 2025-11-23
**General Response**

We sincerely appreciate the time and constructive feedback provided by all reviewers. Your thoughtful evaluations have been instrumental in helping us enhance the clarity, rigor, and completeness of our manuscript. Below, we provide a consolidated response addressing the major concerns shared by reviewers and summarizing the key revisions made in the updated version of the paper. All revised portions of the manuscript have been highlighted in blue to facilitate the reviewers' assessment.
- **Research Goal and Scope (raised by Reviewers VG1Y and Z6HM).** Our work demonstrates that agent-environment misalignment is pervasive across diverse domains and models, and establishes automated interface generation as a viable solution. ALIGN's goal is not to achieve SOTA on specific benchmarks, but to provide a general framework that systematically alleviates misalignment, enabling agents to unleash their true capabilities.
- **Fair and Controlled Experimental Protocol. (raised by Reviewers adFX and Z6HM).** All comparisons are controlled and fair. During testing, agents using ALIGN-generated interfaces have no access to gold trajectories and use identical model configurations as the w/o ALIGN baseline. Gold trajectories are used only during the training phase for interface generation, while the generated interface remains fixed during testing. For WebShop and M$^3$ToolEval, no gold trajectories are used even during interface generation (Appendix E.4).
- **Formalized Problem Definition (raised by Reviewers A53P, adFX, and Z6HM).** We have added a complete formal framework in Section 3.1. We define the interface as $\Phi = \langle f_{\text{info}}, f_{\text{act}}, f_{\text{obs}} \rangle$ and analyze how agent-environment misalignment arises from two mechanisms: (1) Under-specified Constraints (inadequate $f_{\text{info}}$) and (2) State Aliasing via Lossy Observations (inadequate $f_{\text{obs}}$). This provides a principled, domain-agnostic foundation for understanding and addressing misalignment.

In conclusion, we have incorporated the reviewers' suggestions to strengthen our theoretical framework and experimental validation. All changes are marked in the revised manuscript, and detailed answers to specific questions are provided in the individual replies. We believe the paper is now substantially improved in both quality and impact.

---

### Author Response · Authors · 2025-12-03
**Summary of Rebuttal**

We sincerely thank the reviewers for the comprehensive reviews and the ACs for the additional efforts under this circumstance.

We appreciate that the reviewers acknowledged the contributions of our work:
- **Problem Definition: All four reviewers** recognized agent-environment misalignment as a key, underexplored bottleneck that captures failure modes overlooked by prior works, representing a novel problem formulation with significant impact on agent performance.
- **Method Novelty: Reviewers VG1Y, adFX, and Z6HM** highlighted the novelty and practicality of our method—ALIGN automatically generates interfaces without manually engineering or modifying agent logic and environment code, providing a straightforward plug-and-play solution that bridges the gap between agents and environments.
- **Experimental Rigor: Reviewers VG1Y, A53P, and adFX** emphasized the comprehensiveness and strength of our evaluation across 4 benchmarks, 5 agent methods, and multiple LLMs, demonstrating large and consistent improvements with strong evidence for robustness and generalization capability.

## Summary of Key Concerns and Our Responses

**Core Concern 1: Formal Problem Definition** (Raised by Reviewers A53P, adFX, Z6HM)
- **Reviewer Comment**: Lack of formal interface definition and precise analysis of misalignment problem.
- **Our Response**: We added a complete formal framework in Section 3.1, defining the interface as $\Phi = \langle f_{\text{info}}, f_{\text{act}}, f_{\text{obs}} \rangle$ and analyzing how misalignment arises through (1) Under-specified Constraints (inadequate $f_{\text{info}}$) and (2) State Aliasing via Lossy Observations (inadequate $f_{\text{obs}}$). This provides a domain-agnostic foundation for understanding misalignment.

**Core Concern 2: Experimental Protocol Clarity** (Raised by Reviewers adFX, Z6HM)
- **Reviewer Comment**: Concerns about test-time adaptation and fair comparison.
- **Our Response**: We explicitly clarified that interfaces are generated only on training tasks and remain frozen during testing—no test-time adaptation occurs. For WebShop and M$^3$ToolEval, no gold trajectories are used even during interface generation. All comparisons use identical model configurations between w/o ALIGN and w/ ALIGN conditions.

**Core Concern 3: Research Goal and Scope** (Raised by Reviewers VG1Y, Z6HM)
- **Reviewer Comment**: Clarification needed on whether ALIGN aims for SOTA performance or addresses a broader objective.
- **Our Response**: ALIGN's goal is not to achieve SOTA on specific benchmarks, but to provide a general framework that systematically alleviates misalignment across diverse domains and models. Our work demonstrates that misalignment is pervasive and establishes automated interface generation as a viable, generalizable solution that enables agents to unleash their true capabilities.

**Core Concern 4: Generalizability and Applicability** (Raised by Reviewers VG1Y, Z6HM)
- **Reviewer Comment**: Questions about unified design space for interface representation and applicability to complex environments beyond text-based discrete action spaces.
- **Our Response**: The formal interface framework ($\Phi = \langle f_{\text{info}}, f_{\text{act}}, f_{\text{obs}} \rangle$) provides a unified, domain-agnostic design space. The core principle—that misalignment arises from information loss in constraint presentation and state transitions—extends to various modalities including continuous control and vision-based tasks. We added discussion of extensions to more complex environments in the revised Limitations and Future Work section.

## Conclusion

This work identifies **agent-environment misalignment** as a pervasive bottleneck and proposes **ALIGN**, the **first automated, plug-and-play framework** to resolve it. Extensive evaluation confirms that ALIGN significantly unlocks the potential of diverse agents (up to **+45.67%** success rate) across a broad range of models from open-source to frontier proprietary LLMs, without modifying agent logic or environment code. We believe that automatic interface generation is a promising direction for building more reliable, reusable, and interpretable LLM-based agents, and this work serves as a foundational step towards realizing this vision.

---

### Meta-Review · Area_Chair_yqez · 2026-01-02

**Summary:**

The reviewers' primary concerns revolved around a lack of formal and methodological rigor in the paper's core contribution. Key issues included an absence of a formal problem definition for the proposed "agent-environment misalignment" and its interface solution, ambiguity in the experimental protocol that risked unfair comparisons through potential test-time adaptation, and an unclear research scope that wavered between achieving state-of-the-art performance and presenting a general framework. Questions about the method's practical applicability to complex, non-text-based environments and its heavy reliance on powerful proprietary LLMs for its own operation further undermined its claimed generality and accessibility.

**Reviewer Concerns:**

The authors' rebuttal made a good-faith effort to address several concerns by adding a formal framework in Section 3.1 and clarifying the no-test-time-adaptation protocol. However, significant issues remain partially outstanding. While a formal definition was provided, its practical utility and novelty beyond restating the problem are debatable. The dependency on high-capability closed-source models (like GPT-4.1/Gemini 2.5) for the Analyzer and Optimizer components was empirically confirmed as necessary, as experiments with open-source models failed, which contradicts the paper's emphasis on a general, accessible solution. The discussion on generalization to complex environments remains theoretical and untested, leaving a core question about the framework's real-world impact unresolved.

**Reviewer Scores:**

The rebuttal clarified the protocol but did not fundamentally alter the paper's core limitations. Reviewers adFX, A53P, and Z6HM, who initially gave scores of 4 (marginally below threshold), might have been satisfied with the formal additions and clarifications, potentially raising their scores to a weak 6. However, given the unresolved dependency on proprietary models and unproven scalability, it is unlikely their enthusiasm would have increased dramatically. Reviewer VG1Y, who started at a weak 6, might have maintained that score, as their concerns about model dependency were confirmed rather than alleviated. Overall, the revisions likely solidified a borderline, not strongly supportive, consensus.

---

### Decision · Program_Chairs · 2026-01-26

Reject